# The neuropeptide tachykinin is essential for pheromone detection in a gustatory neural circuit

**Shruti Shankar[1,2], Jia Yi Chua[1], Kah Junn Tan[1], Meredith EK Calvert[1†], Ruifen Weng[3], Wan Chin Ng[1], Kenji Mori[4], Joanne Y Yew[1,2,5*‡]**

[1]Temasek Life Sciences Laboratory, Singapore, Singapore; [2]Department of Biological Sciences, National University of Singapore, Singapore, Singapore; [3]Institute of Molecular and Cell Biology, Singapore, Singapore; [4]Photosensitive Materials Research Center, Toyo Gosei Co., Ltd, Chiba, Japan; [5]Pacific Biosciences Research Center, University of Hawaii at Mānoa, Honolulu, United States

**Abstract** Gustatory pheromones play an essential role in shaping the behavior of many organisms. However, little is known about the processing of taste pheromones in higher order brain centers. Here, we describe a male-specific gustatory circuit in *Drosophila* that underlies the detection of the anti-aphrodisiac pheromone (3R,11Z,19Z)-3-acetoxy-11,19-octacosadien-1-ol (CH503). Using behavioral analysis, genetic manipulation, and live calcium imaging, we show that Gr68a-expressing neurons on the forelegs of male flies exhibit a sexually dimorphic physiological response to the pheromone and relay information to the central brain via peptidergic neurons. The release of tachykinin from 8 to 10 cells within the subesophageal zone is required for the pheromone-triggered courtship suppression. Taken together, this work describes a neuropeptide-modulated central brain circuit that underlies the programmed behavioral response to a gustatory sex pheromone. These results will allow further examination of the molecular basis by which innate behaviors are modulated by gustatory cues and physiological state.

*For correspondence: jyew@hawaii.edu

**Present address:** †Gladstone Histology and Light Microscopy Core, The J. David Gladstone Institutes, University of California, San Francisco, San Francisco, United States; ‡Pacific Biosciences Research Center, University of Hawaii at Mānoa, Honolulu, United States

**Competing interests:** The authors declare that no competing interests exist.

## Introduction

For many animals, exogenously released chemical cues known as pheromones heavily influence social behaviors that are crucial to survival and reproduction (*Karlson and Luscher, 1959*). Elucidating the neural basis of pheromone detection provides a means for understanding how information from sensory stimuli is encoded and used to modulate complex behaviors such as mating (*Pavlou and Goodwin, 2013*; *Haga-Yamanaka et al., 2014*) and aggression (*Chamero et al., 2007*; *Wang et al., 2011*; *Fernández and Kravitz, 2013*).

The neural circuits underlying olfactory pheromone detection are well described in the silkmoth *Bombyx mori* (*Sakurai et al., 2014*) and honey bee (*Roussel et al., 2014*). In *Drosophila*, the pathways mediating detection of the sex pheromone 11-*cis*-vaccenyl acetate (cVA) have been refined down to 4 neurons connected by 3 synapses (*Ruta et al., 2010*). In addition, the projection patterns for several other olfactory receptor neurons that likely detect sex pheromones have been mapped from the antennal lobe to the ventral lateral horn, implicating this region in the central brain as a specialized site for processing pheromone odors (*Jefferis et al., 2007*).

In contrast to the olfactory system, the higher order pathways for the processing of gustatory pheromones are largely unknown despite their importance in behavior (*Wang et al., 2011*; *Fernández and Kravitz, 2013*). Several pheromone receptors located in the primary gustatory organs (proboscis, labellum, forelegs) have been identified. Gr32a (*Miyamoto and Amrein, 2008*) is thought

**eLife digest** In many species of animals, the male decides to pursue a potential female mate based on how she smells and tastes. Powerful chemical signals known as pheromones control this decision. When a male fruit fly mates with a female fruit fly, he often leaves behind an anti-aphrodisiac pheromone that, when males taste it, deters them from mating with the female. Until recently, however, little was known about how the brain processes information from such taste pheromones.

Now, Shankar et al. have investigated this problem in a series of experiments with normal and genetically modified flies. In the first experiment normal male flies were exposed to the chemical on its own, to the chemical on a sample of female skin, and to the chemical on actual female flies. The male flies did not respond to the pheromone on its own, but they did respond to it in the other two scenarios.

Next, Shankar et al. used genetic techniques to eliminate individual neurons in the male flies and then observed how the loss of specific neurons influenced the response of the fly to the pheromone. These experiments showed that male flies have a special group of sensory neurons in their legs that detect the chemical and then send an electrical signal to the brain. Shankar et al. then went on to identify a brain circuit consisting of 8–10 neurons that responds to this signal and to show that the release of a neurochemical called Tachykinin is essential in communicating the signal.

In a final set of experiments, Shankar et al. introduced two sensors—one in the sensory neurons in the legs, the other in the 8–10 neurons in the brain—that light up when the neurons in that region are close enough to each other to form connections. The results suggest that the sensory neurons in the legs form connections with the 8–10 neurons in the brain.

A challenge for the future is to understand how the nervous system combines different social cues and information about the physiological state of the animal, and how this influences the decision to mate.

to respond to the male pheromone (7Z)-tricosene (*Wang et al., 2011*; *Fan et al., 2013*; *Andrews et al., 2014*). Furthermore, Gr33a (*Moon et al., 2009*), Gr39a (*Watanabe et al., 2011*), and a member of the class of ionotropic receptors (IRs), Ir20a (*Koh et al., 2014*), contribute to courtship behavior, though the ligands remain unidentified. Lastly, a recently discovered class of ion channels belonging to the pickpocket family of proteins (ppk23, ppk25, and ppk29) responds to both female pheromones and male anti-aphrodisiacs (*Thistle et al., 2012*; *Toda et al., 2012*; *Mast et al., 2014*; *Vijayan et al., 2014*). Without exception, processes from all Gr-expressing neurons map to the ventral cord and the subesophageal zone (SEZ) (*Wang et al., 2004*; *Kwon et al., 2014*). However, little is known about the post-synaptic targets of the SEZ as well as the neurotransmitter systems used to mediate pheromone-related behaviors in the central brain.

In this work, we describe a central neural circuit that mediates the detection and behavioral response to a gustatory sex pheromone, (3R,11Z,19Z)-3-acetoxy-11,19-octacosadien-1-ol (CH503). CH503 is transferred from males to females during mating and inhibits courtship from other males (*Yew et al., 2009*). The pheromone also functions as a potent suppressor of male courtship behavior in other drosophilids (*Ng et al., 2014*). We show that CH503 is detected by Gr68a-expressing gustatory neurons on the male foreleg and that information is transduced via peptidergic cells to the central brain. Specifically, a cluster of 8–10 neurons within the SEZ mediates the pheromone-triggered courtship-avoidance response through the release of the neuropeptide tachykinin.

## Results

### CH503 inhibits male courtship behavior in a dose-dependent manner

Courtship in *Drosophila* consists of a stereotyped sequence of behaviors including orientation, wing vibration, tapping with the forelegs, abdomen curling, and copulation (*Spieth, 1974*) (*Figure 1A*). To determine the sensitivity of males to CH503 and to examine the courtship features that are suppressed by the pheromone, wild-type CantonS male flies were placed with virgin females perfumed with doses of synthetic CH503 ranging from 0 to 2667 ng. Males exhibited

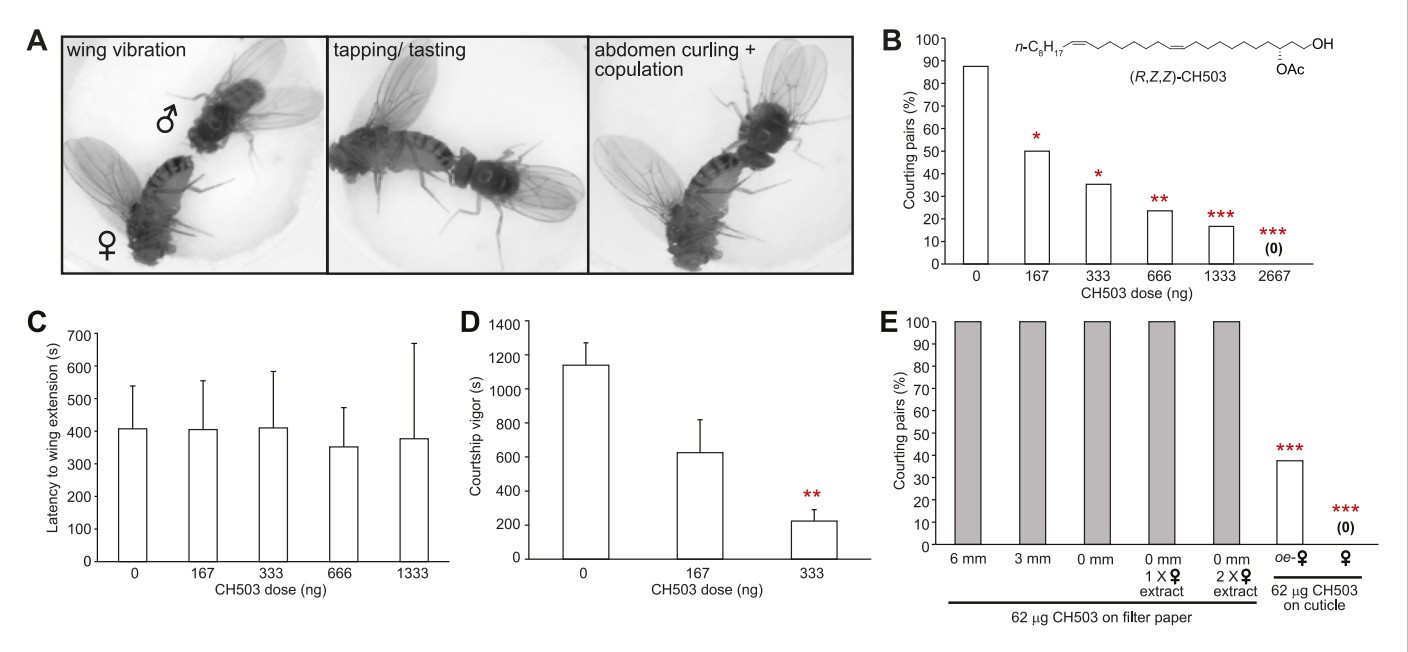

**Figure 1**. Functional properties of the male sex pheromone CH503. (**A**) The typical courtship sequence of *D. melanogaster* is comprised of wing vibration performed by the male towards the female, tapping and tasting of the female abdomen with the forelegs, and abdomen curling followed by copulation. (**B**) Courtship behavior exhibited by wild-type Drosophila males decreases in a dose-dependent manner with increasing amounts of CH503 on the surface of virgin females. N = 25–30, Fisher's exact probability test, *p < 0.05, **p < 0.01, ***p < 0.001. (**C**) CH503 does not change the latency to courtship initiation, as measured by the latency to wing vibration. N = 16–24, ANOVA with Tukey's multiple comparison test. Error bars represent SEM. (**D**) CH503 suppresses the amount of time the fly actively spends courting. Courtship vigor is defined as the total time the male spends courting, calculated from the first instance of orientation and wing vibration. N = 8–24, ANOVA with Tukey's multiple comparison test, **p < 0.01. Error bars represent SEM. (**E**) CH503 has low volatility and inhibits courtship only when detected on female cuticles. The absence of female cuticular hydrocarbons in oenocyte-less (*oe-*) flies also did not affect CH503-induced courtship suppression. N = 8, Fisher's exact probability test, ***p < 0.001.

a dose-dependent response to the pheromone. A dose of 167 ng/fly was sufficient to suppress courtship in approximately 50% of the behavioral trials (*Figure 1B*). The latency to courtship initiation (as measured by the first instance of wing vibration) was similar across all doses, ranging from 350 to 400 s (*Figure 1C*). However, the overall courtship vigor was significantly suppressed by the pheromone (*Figure 1D*). Taken together, these findings indicate that CH503 inhibits later stages of the courtship sequence and sustained courtship behavior.

## CH503 is a low volatility contact cue and is effective only when detected on female cuticles

Wing extension is one of the first steps of the courtship ritual and can be triggered in the absence of tactile contact through visual cues (*Ejima and Griffith, 2008*; *Pan et al., 2012*; *Agrawal et al., 2014*) and volatile pheromones (*Tompkins et al., 1980*; *Venard and Jallon, 1980*). Since the latency to wing extension did not appear to be affected by CH503, we hypothesized that the pheromone is detected only at close proximity. To determine whether tactile contact with CH503 is necessary for courtship suppression, we positioned a pheromone source separated from the female target at various distances. To prevent contact with the pheromone, a mesh barrier was placed between the courtship chamber and a second chamber containing a piece of filter paper soaked with 64 μg of CH503. Despite the high dose of pheromone, male courtship was uninhibited when the filter paper was placed 6 or 3 mm away, or on the floor of the courtship chamber (allowing for direct contact). In each case, 100% of male flies initiated courtship towards females (*Figure 1E*). Potentially, female pheromones act in synergy with CH503 and both cues are needed to inhibit courtship. However, a mixture of female cuticular extract and CH503 applied to filter paper that was placed on the floor of the chamber was ineffective (*Figure 1E*). We further tested for synergistic effects from female

pheromones by using transgenic female flies in which oenocytes, the pheromone-producing cells in *Drosophila*, were genetically ablated. In the presence of oenocyte-less flies perfumed with CH503, male courtship was still significantly inhibited (*Figure 1E*). In sum, these results indicate that female-specific pheromones synthesized in the oenocytes do not mediate the detection of CH503. Furthermore, CH503 is effective as an anti-aphrodisiac only when placed on the cuticular surface of females, indicating that sensory cues other than cuticular lipids are required.

## CH503 is detected by taste, not smell

Based on the relatively high molecular weight of CH503 and its low volatility, we hypothesized that CH503 is likely to be perceived as a tastant. We performed a proboscis extension reflex (PER) assay to test whether CH503 is detected by foreleg gustatory receptors. Stimulation of the foreleg with rewarding stimulants such as a sugar solution induces extension of the proboscis (*Kimura et al., 1986*) while aversive, bitter substances suppress the PER (*Lacaille et al., 2007*). Exposure of the forelegs to 0.5–2.0 µg of CH503 together with 4% sucrose reduced the frequency of the PER (*Figure 2A*). Similarly, caffeine, a bitter stimulus, also suppressed the PER. In contrast, females did not exhibit a change in the PER when exposed to CH503, indicating that either the pheromone was not detected or that it does not carry a negative valence for females (*Figure 2—figure supplement 1*). To further examine whether CH503 is detected as an odorant or tastant, we tested the responses of transgenic males defective in smell or taste perception. Flies lacking the olfactory co-receptor Or83b (*Orco*) exhibit significant defects in olfactory pheromone detection (*Larsson et al., 2004*). However, Or83b-defective flies still suppressed courtship in the presence of CH503 (*Figure 2B*). In contrast, *Voila*[1] mutants, which exhibit gustatory defects (*Balakireva et al., 1998*), continued to court CH503-perfumed females (*Figure 2B*). Taken together, these results indicate that CH503 is perceived as an aversive tastant by males and not females and is detected by gustatory receptors on the foreleg.

## CH503 is detected by Gr68a neurons on the forelegs of male flies

To identify the subset of neurons on the male foreleg that detect CH503, we performed a behavioral screen by using the *Gal4-UAS* transgene system to ablate or functionally suppress each of the 19 known foreleg-specific gustatory neurons (*Ling et al., 2014*). The courtship behavior of transgenic or mutant males was then tested in the presence of CH503-perfumed females. The synthetic stereoisomer (*S, Z, Z*)-CH503 (*Figure 2—figure supplement 2*) was used for preliminary screening since it is more potent than naturally occurring (*R, Z, Z*)-CH503 both in suppressing courtship (*Shikichi et al., 2012*; *Ng et al., 2014*) and the PER (*Figure 2—figure supplement 1*) and thus, would lessen the likelihood of false positives. Males from 15 of the gustatory *Gal4* lines displayed normal courtship suppression behavior in the presence of CH503-perfumed females (*Figure 2B*). Four lines exhibited significantly reduced levels of courtship behavior with control females, thus confounding our ability to detect CH503-related courtship suppression. However, males in which the Gr68a-encoding gene was transcriptionally silenced continued to court females in the presence of (*S, Z, Z*)-CH503 (*Figure 2B*). A similar response was found when males were tested with the natural pheromone at a 333 ng/fly dose using two independent *Gal4* lines (*Figure 2C*). To ensure complete loss of Gr68a expression, we generated a *ΔGr68a* mutant using targeted ends-out homologous recombination and confirmed the loss of expression with PCR (*Figure 2D*; *Figure 2—figure supplement 3*). *ΔGr68* mutants displayed robust courtship behavior in the presence of CH503-perfumed females (*Figure 2C*). Importantly, the sensitivity to CH503 was restored upon re-introduction of the *Gr68a*-encoding gene (*Gr68a*[Res]; *Figure 2C*). The coding region for *Gr68a* resides within the intronic region of another gene, *CG6024* (*Figure 2D*). However, *CG6024* expression was not significantly changed in either mutant or rescue lines (*Figure 2—figure supplement 3*).

To determine whether activation of Gr68a neurons was sufficient to induce the courtship avoidance response, we expressed the conditionally activated cation channel *Drosophila* TrpA1 (dTrpA1). In the presence of unperfumed female targets, a slight but non-significant suppression in courtship behavior was observed at the activation temperature of 29°C compared with the inactive condition at 19°C (*Figure 2C*). Thus, activation of Gr68a neurons is not sufficient to suppress courtship, possibly due to conflicting signals resulting from mutual activation of mechanosensory and chemosensory neurons (see 'Discussion').

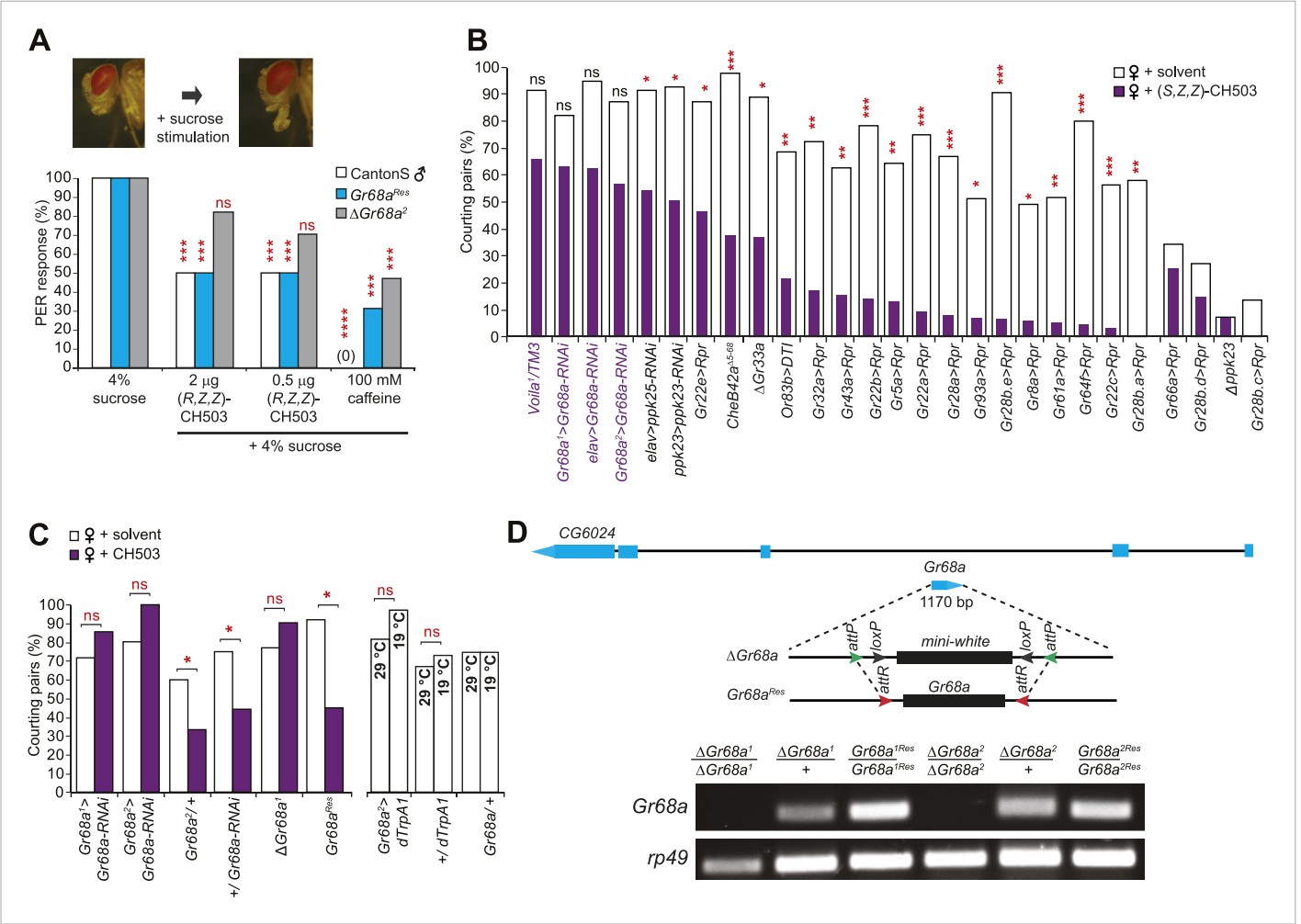

Figure 2. Gr68a expression in the male foreleg is required for CH503 detection. (A) Simultaneous stimulation of the male foreleg with 4% sucrose and CH503 or caffeine significantly inhibits the proboscis extension reflex (PER; shown in pictures) in CantonS males (white). The PER suppression was not observed in ΔGr68a mutant flies (gray) but was restored upon re-introduction of the Gr68a gene (Gr68a^Res; blue). For each genotype, the response to each test compound was compared to the response to sucrose alone. N = 18, Fisher's exact probability test, ***p < 0.001, ****p < 0.0001, ns: not significant. (B) A behavioral screen targeting foreleg-specific gustatory receptor neurons, pheromone receptors, and a pheromone binding protein reveals that Gr68a is a candidate receptor for detecting (S, Z, Z)-CH503. The number of flies exhibiting courtship in response to the pheromone (purple) was compared to the response to a solvent-perfumed female (white). For some genotypes (far right of graph), the basal courtship level was too low to observe a courtship suppression effect. N = 12–73, Fisher's exact probability test, *p < 0.05, **p < 0.01; ***p < 0.001. (C) Silencing Gr68a expression with RNAi or genetic deletion resulted in a loss of sensitivity to CH503. The courtship suppression response was unaltered in parental control lines and restored upon re-introduction of the gene into the mutant. Hyperactivation of Gr68a-expressing neurons using dTrpA1 at the activation temperature (29°C) resulted in a slight but non-significant courtship suppression compared to the inactive temperature (19°C). Parental control lines exhibited no difference in courtship behavior at 29°C or 19°C. N = 12–37, Fisher's exact probability test, *p < 0.05, ns: not significant. (D) (Top) A schematic of the Gr68a gene locus shows that the single coding exon (blue) resides within the intronic region (black) of another gene, CG6024. (middle) The homologous recombination strategy for deletion and rescue of Gr68a involves replacement of the endogenous gene with the mini-white marker using recombinase-mediated cassette exchange (RMCE). Genomic rescue of Gr68a is accomplished by exchanging mini-white via RMCE with the Gr68a sequence. (Bottom) Analysis by semi-quantitative PCR of genomic DNA shows the complete absence of Gr68a expression in two homozygous mutant alleles and successful rescue in the respective Gr68a^Res lines. Rp49 expression is used as a loading control. CG6024 expression is not changed in mutant or rescue lines (Figure 2—figure supplement 3).

The following figure supplements are available for figure 2:

Figure supplement 1. Sexually dimorphic PER response to CH503.

Figure supplement 2. Chemical structures of (R, Z, Z)-CH503, (S, Z, Z)-CH503, and CH503 analogs.

Figure supplement 3. Characterization of ΔGr68a mutant alleles by quantitative PCR.

## Gr68a neurons exhibit a dose-dependent physiological response to CH503

To visualize *Gr68a* neurons and their processes, we drove expression of a membrane-tethered green fluorescent protein molecule (*UAS-mCD8::GFP*) using *Gr68a-Gal4*. Consistent with previous reports (*Bray and Amrein, 2003*; *Ejima and Griffith, 2008*; *Ling et al., 2014*), a sexually dimorphic GFP expression pattern was observed in the chemosensory neurons of gustatory bristles found in male forelegs (*Figure 3A*). Each of the male tarsal segments exhibited more labeled neurons than females (*Table 1*). Labeled non-neuronal cells (distinguished by larger, irregularly shaped membranes lacking projections) were also observed but only in male legs (*Figure 3A*; *Table 1*). Notably, knockdown of Gr68a expression using the pan-neuronal *elav-Gal4* driver resulted in a loss of sensitivity to CH503 (*Figure 2B*). The results recapitulate the phenotype observed with *Gr68a-Gal4* and indicate that neuronally expressed receptors are important for CH503 detection.

To measure directly the response of Gr68a-expressing neurons to CH503, we performed in vivo calcium imaging of the forelegs of live flies by expressing the calcium sensor GCaMP5 under control of *Gr68a-Gal4*. The maximum fluorescent change ($\Delta F/F$) was elicited with a bath-applied dose of 500 ng of (*R*, *Z*, *Z*)-CH503, close to the minimum dosage required to elicit a behavioral response in courtship assays (*Figure 3B*). At this dose, 6 of 9 cells in segments T2–4 showed significant responses compared to the control solvent, ranging from $\Delta F/F$ 0.34 $\pm$ 0.05 to 1.39 $\pm$ 0.46 (mean $\pm$ SEM). At a dose of 50 ng, only 1 cell (T4, Cell a) showed a significant signal increase, with an average $\Delta F/F$ of 1.52 $\pm$ 0.37. Gr68-neurons exhibited two general patterns of responses: more commonly, a phasic response is observed where maximum fluorescent intensity occurred immediately after the addition of the pheromone and was followed by a gradual decline to baseline levels after 6 s. A tonic response is also seen where fluorescence gradually increased and peaked after approximately 120 s (*Figure 3C*; *Figure 3—figure supplement 1*; *Video 1*). Gr68a-expressing neurons also responded to the synthetic stereoisomer (*S*, *Z*, *Z*)-CH503 over the same range of doses though they differed in terms of dynamic range and response pattern (*Figure 3—figure supplement 2*).

Stimulation of Gr68a-expressing neurons with a synthetic CH503 analog containing two triple bonds did not elicit a change in fluorescent signal (*Figure 3B*), consistent with previous observations that the analog is behaviorally inert (*Shikichi et al., 2013*). Additionally, CH503 failed to evoke a physiological response when Gr68a expression was either reduced using RNAi or eliminated in *ΔGr68a* mutants (*Figure 3B*; *Figure 3—figure supplement 3*). In females, the responses to the pheromone from the forelegs were indistinguishable from that of the solvent control (*Figure 3D*). In the case of cells j and s, it may be that the responses are too subtle to be reliably measured without a sample size ~100. The functional imaging results are consistent with observations from the appetitive PER assay where females also failed to exhibit a measurable response to CH503 (*Figure 2—figure supplement 1*). Taken together, direct measurement of neural response in Gr68a-expressing neurons reveals robust functional activation by CH503 in males only. Moreover, the double-bonds in the carbon backbone of the CH503 molecule are an essential structural feature for pheromone activity.

## The role of pickpocket neurons in CH503 detection

Recently, male and female leg neurons expressing the ion channel ppk23 were shown to respond physiologically to non-volatile gustatory pheromones in *D. melanogaster* (*Lu et al., 2012*; *Thistle et al., 2012*; *Toda et al., 2012*). We measured responses from GCaMP-expressing *ppk23-Gal4*-labeled cells in the labella and forelegs of both male and female flies. In the male proboscis, CH503 activated approximately 8 of 14 neurons while none of the female ppk23 neurons responded (*Figure 3—figure supplement 4*; *Video 2*). The response of ppk23 neurons in the male foreleg was more heterogeneous compared to Gr68a-expressing neurons. Depending on the cell, neural responses could be elicited by the natural pheromone (*R*, *Z*, *Z*)-CH503, the artificial stereoisomer (*S*, *Z*, *Z*)-CH503, a synthetic CH503 analog or solvent controls (*Figure 3—figure supplement 5*). While most ppk23-expressing neurons typically responded in a phasic manner with a maximum change in $\Delta F/F$ shortly after pheromone addition, some cells exhibited bursting responses with large increases in $\Delta F/F$ that recurred frequently during the time course of the experiment (*Figure 3—figure supplement 5*; *Video 3*).

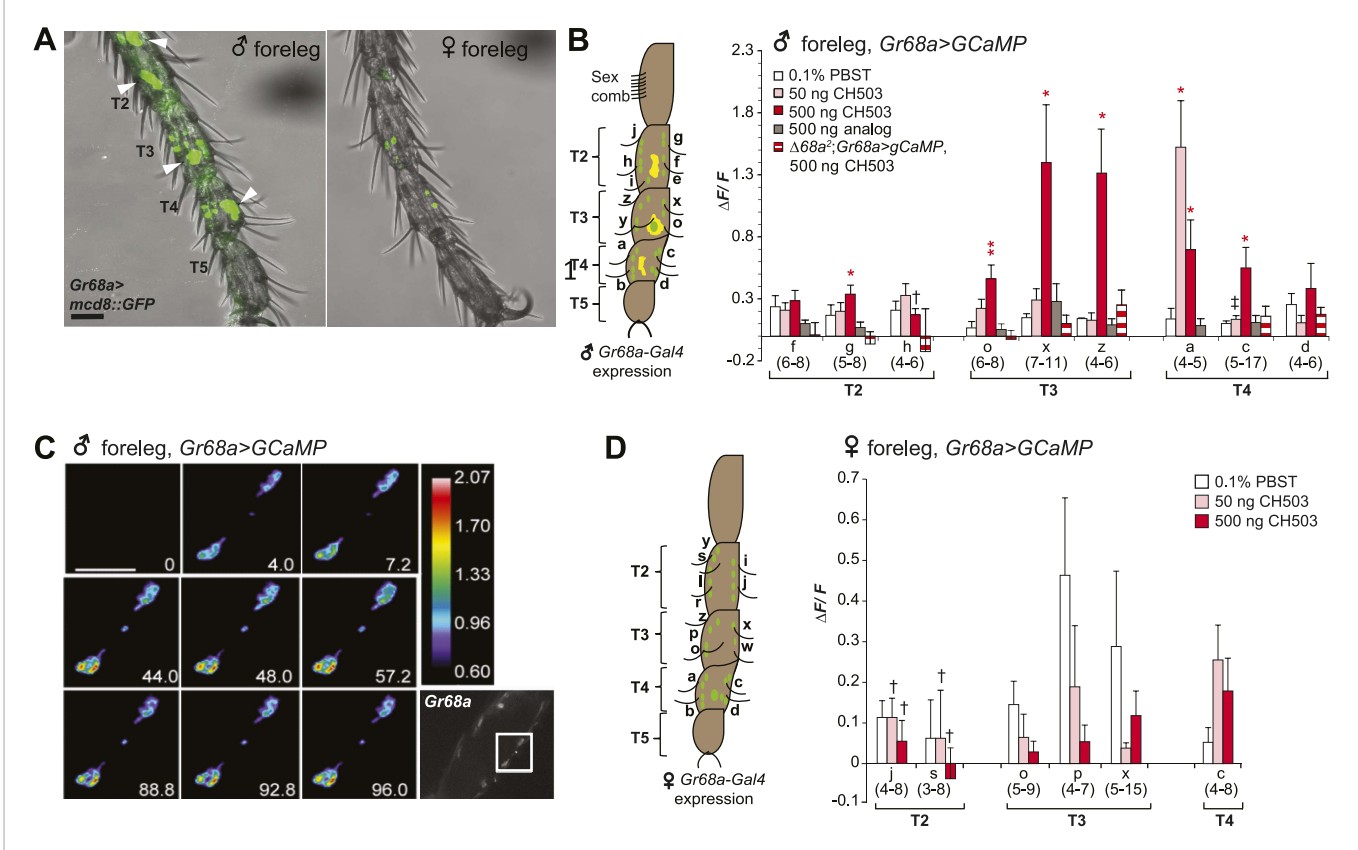

**Figure 3**. Gr68a is essential for CH503-evoked neuronal responses in the male foreleg. (**A**) Visualization of GFP-labeled Gr68a-expressing neurons reveals neuronal and non-neuronal cells (arrowheads) in tarsal segments T2-5 from the male foreleg. Scale bar: 35 µm. (**B**) Gr68a-expressing neurons in the male foreleg show changes in Ca$^{2+}$ activity in response to two doses of CH503 (pink, red). The behaviorally inert analog (R)-3-Acetoxy-11, 19-octacosadiyn-1-ol fails to elicit a significant response (gray). No increase in ΔF/F is observed from the forelegs of ΔGr68a-mutant flies (red stripes). Cells are designated according to the schematic (left) showing sensory neurons (green) and non-neural cells (yellow). For each cell type, the averaged response ± SEM and sample size is shown; Student's *t*-test with unequal variance, *p < 0.05, **p < 0.01. Unless otherwise indicated, statistical power is at least 0.8 for a significance level of 0.05 for the 50 and 500 ng CH503 doses. ‡N = 47 required for power of 0.8; †N = 201 required for 0.8 power. (**C**) A color-coded time course from 0–96 s showing the response in T2 Gr68a neurons evoked by 500 ng of CH503. The positions of the neurons on the foreleg are shown in the raw fluorescent image (bottom right corner, square). See also ***Figure 3—figure supplement 1*** and ***Video 1***. Scale bar: 10 µm. (**D**) Gr68a-expressing neurons on the female foreleg do not show a statistically significant response to (R, Z, Z)-CH503. Student's *t*-test with unequal variance, p > 0.05 for all cells tested. Error bars indicate SEM; sample sizes are shown below each cell type. Unless otherwise indicated, statistical power is at least 0.8 for a significance level of 0.05 for the 50 and 500 ng CH503 doses. †N ~ 100 needed to achieve 0.8 power.

The following figure supplements are available for figure 3:

**Figure supplement 1**. Line graph representation showing the tonic response of a T2 Gr68a neuron upon stimulation with 500 ng of CH503.

**Figure supplement 2**. Physiological responses of male Gr68a neurons to (S, Z, Z)-CH503.

**Figure supplement 3**. Physiological responses of Gr68a neurons upon RNAi-mediated silencing of *Gr68a* expression.

**Figure supplement 4**. Physiological responses of ppk23 proboscis neurons to (R, Z, Z)-CH503.

**Figure supplement 5**. Physiological responses of ppk23 leg neurons to (R, Z, Z)-CH503.

**Figure supplement 6**. *Gr68a-Gal4* and *fruitless (fru)*-expression in the foreleg do not co-localize.

**Table 1**. Average number of GFP-positive cells in male and female foreleg segments labeled using *Gr68a-Gal4*

|  | T1 | T2 | T3 | T4 | T5 | Total* |
|---|---|---|---|---|---|---|
| ♂ neurons | 2 ± 1 | 2 ± 1 | 3 ± 1 | 2 ± 1 | 0 (9/12 flies) | 9 ± 1 |
| ♀ neurons | 2 ± 1 | 1 ± 1 | 2 ± 1 | 1 ± 1 | 0 (11/12 flies) | 6 ± 2 |
| ♂ non-neural cells | 2 ± 1 | 3 ± 0 | 2 ± 0 | 1 ± 0 | 0 | 8 ± 1 |
| ♀ non-neural cells | 0 | 0 | 0 | 0 | 0 | 0 |

*Averaged count (±SD) from 12 flies.

To test whether ppk23 plays a role in mediating the behavioral response to CH503, we examined the behavior of male flies in which *ppk23* expression was silenced using RNAi. Males continued to respond to CH503 though the courtship suppression effect is partially mitigated compared to other gustatory receptors (*Figure 2B*). It was not possible to assess the courtship behavior of Δ*ppk23* mutants due to low basal courtship levels. Knockdown of a second member of the ppk family, ppk25, also resulted in a partially reduced sensitivity to CH503 (*Figure 2B*). While these results do not eliminate the possibility that ppk-23/25-expressing neurons contribute to CH503 detection, both the physiological and behavioral data indicate that the responses of the cells by themselves are not the primary sensory pathway and do not appear to be necessary for courtship suppression to occur.

## Identification of higher order neurons involved in CH503 detection

The *Gr68a-Gal4*-labeled neurons send axonal projections to each of the six thoracico-abdominal (TAG) neuromeres and extend into the SEZ and the antennal mechanosensory and motor center (AMMC) (*Figure 4A*). To identify upstream pathways that relay information from the SEZ, we screened 22 *Gal4* drivers and 9 *UAS-RNAi* lines targeting central brain regions, neuropeptide systems, and neurotransmitter circuits for their contribution to CH503-induced courtship suppression (*Figure 4—figure supplement 1*; *Table 2*). Interestingly, suppression of neural activity using the *c929-Gal4* line resulted in a marked decrease in sensitivity to CH503 (*Figure 4B*). The peptide driver labels both neuroendocrine and peptidergic neurons in the central nervous system and processes in the TAG ([*Taghert et al., 2001*; *Hewes et al., 2003*]; *Figure 4—figure supplement 2*). We partially limited *c929-Gal4* expression by suppressing Gal4 activity in the ventral nerve cord with the repressor *tsh-Gal80* and observed that sensitivity to CH503 was not restored (*Figure 4B*; *Figure 4—figure supplement 2*). Hence, *c929*-labeled processes in the thoracic ganglia do not appear to contribute to the behavioral response to CH503.

We next attempted to identify other neurotransmitters within the *c929-Gal4* circuit that could mediate pheromone processing. RNAi-mediated silencing of genes related to aminergic, cholinergic, glutamatergic, and GABAergic synthesis, transport, and receptor systems did not have an effect (*Figure 4—figure supplement 3*). However, genetic ablation or suppression of neural activity in two neuropeptidergic circuits, neuropeptide F (NPF) and tachykinin (TK), reduced sensitivity to CH503 (*Figure 4C*; *Figure 5A*).

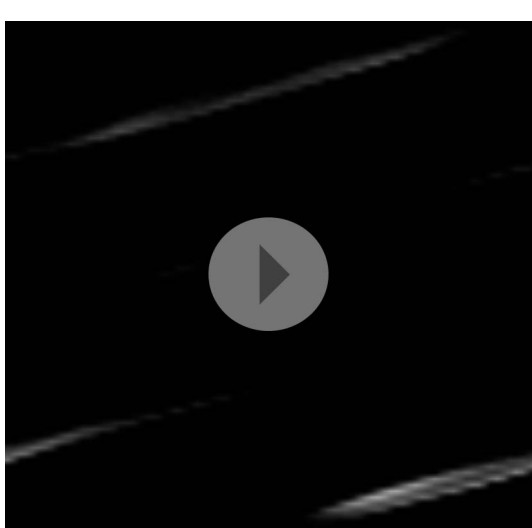

**Video 1.** Physiological response from Gr68a neurons on the male foreleg expressing GCaMP. Cell bodies in T3 exhibit a tonic response upon stimulation with 500 ng of CH503.

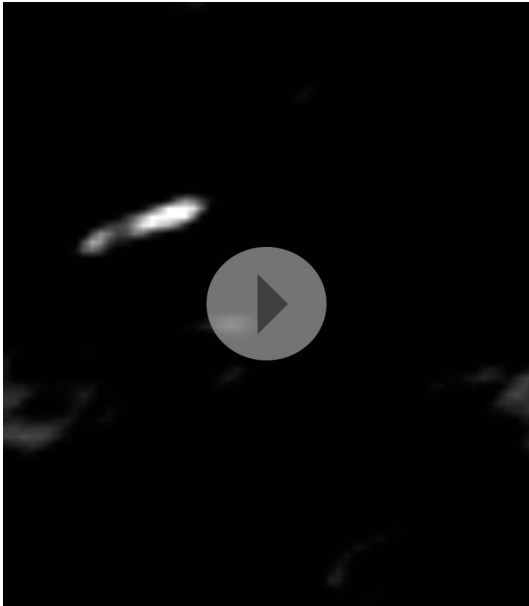

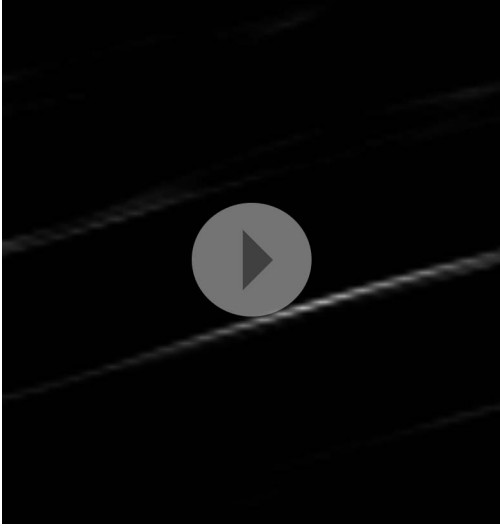

**Video 2.** Physiological response from ppk23 neurons on the male proboscis expressing GCaMP. Following stimulation with 500 ng of CH503, the projections of ppk23 neurons exhibit a bursting response. The cell bodies respond in a tonic manner, displaying a gradual increase in fluorescence intensity. The tonic response could be due to persistent stimulation from the pheromone.

**Video 3.** Physiological response from ppk23 neurons on the male foreleg expressing GCaMP. Following stimulation with 500 ng of CH503, cell bodies and projections in T3 display a bursting response.

The peptide NPF co-localizes with *c929-Gal4* throughout the TAG, SEZ, and protocerebrum (*Figure 4—figure supplement 2*). Anatomical analysis revealed that Gr68a synaptic terminals and NPF-positive processes are closely apposed to each other in the SEZ, implicating NPF-expressing circuits as a putative second order relay to the central brain (*Figure 4D*). Notably, silencing *NPF* expression using *c929-Gal4* or the pan-neural driver *elav-Gal4* (which caused a 5.3-fold decrease in transcript level, *Figure 4—figure supplement 4*) was ineffective at changing the male inhibitory courtship response (*Figure 4C*). These findings indicate that the NPF peptide itself is not likely to be essential for mediating CH503-related behavior. Consistent with this observation, silencing of the neuronal circuits associated with the NPF receptor, NPFR, also did not change the response to CH503 (*Figure 4—figure supplement 1*).

## TK-expressing cells are a second order circuit for gustatory pheromone information

TK-expressing cells represent a second higher order neural circuit that mediates CH503-induced courtship suppression. We observed a significantly altered response in males both to the natural pheromone (*Figure 5A*) and the more potent (*S, Z, Z*)-CH503 stereoisomer (*Figure 5—figure supplement 1*) following genetic cell ablation or functional suppression of TK-expressing cells. The results were consistent using two independent *TK-Gal4* drivers which label overlapping but not identical cell populations. Genetic excision of the *TK* gene also caused a striking loss of sensitivity to CH503 (*Figure 5A*). The phenotype was rescued upon ectopic expression of *TK* in the mutant background (*Figure 5A*). Thus, in contrast to our findings with the NPF peptide, the product of the *TK* gene is necessary for mediating the behavioral actions induced by CH503.

To examine TK expression relative to Gr68a processes, we used an antibody to TK to label brain tissue from *Gr68a-Gal4>UAS-sybGFP* flies. Positive TK staining is evident in the SEZ and AMMC and is positioned closely to *Gr68a-Gal4*-labeled terminals (*Figure 5B,C,E*). To determine if there is synaptic connectivity, we used the GFP reconstitution across synaptic partners (GRASP) method (*Feinberg et al., 2008*; *Gordon and Scott, 2009*). Complementary fragments of membrane-tethered GFP are

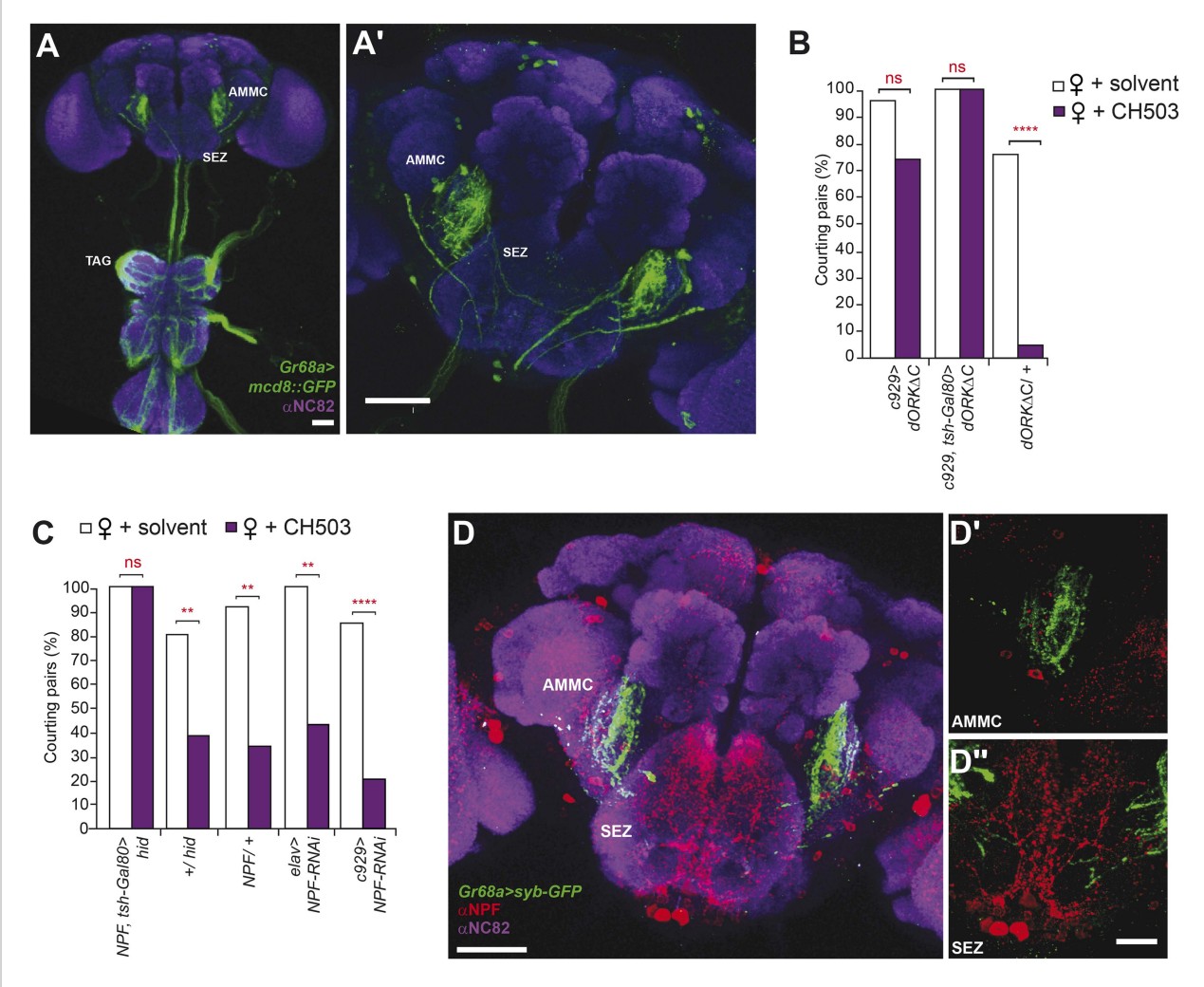

**Figure 4**. Higher order neural circuits essential for processing CH503. (**A**) *Gr68a-Gal4*-labeled afferent projections extend to the thoracico-abdominal ganglia (TAG), subesophageal zone (SEZ), and antennal mechanosensory and motor center (AMMC). Image represents a maximum intensity Z-series projection. Scale bar **A**: 25 μm; **A'**: 50 μm. (**B**) Inhibition of electrical activity in *c929-Gal4*-labeled neurons with *UAS-dORKΔC*, an inwardly rectifying K⁺ channel, resulted in high courtship levels in the presence of CH503. Suppressing *Gal4* expression in the ventral cord with a *tsh-Gal80* transgene (hence, limiting dORKΔC expression primarily to the central brain) failed to restore sensitivity to CH503. No change in CH503 response was observed in the absence of the *c929-Gal4* driver. N = 23–25, Fisher's exact probability test, ns: not significant, ****p < 0.0001. (**C**) Ablation of central brain neural circuits associated with NPF abolished the courtship suppression response to CH503. The courtship behavior of genetic controls was unaffected. Silencing *NPF* expression in all neural cells (using *elav-Gal4*) or in peptidergic neurons (using *c929-Gal4*) did not alter flies' sensitivity to CH503. N = 14–33, Fisher's exact probability test, ns: not significant, **p < 0.01, ****p < 0.0001. (**D**) NPF-expressing processes are closely apposed to *Gr68a-Gal4* synaptic terminals labeled with synaptobrevin-GFP (syb-GFP) in the AMMC (**D'**) and SEZ (**D''**). No co-localization is observed (Pearson's coefficient: 0.01). Image represents a maximum intensity Z-series projection. Scale bar **D**: 50 μm; **D'**, **D''**: 20 μm.

The following figure supplements are available for figure 4:

**Figure supplement 1**. Central brain screen to identify CH503-processing circuits.

**Figure supplement 2**. Co-expression of anti-NPF immunostaining with *c929-Gal4*-directed GFP expression.

**Figure supplement 3**. Screen of tachykinin and small transmitter systems within the *c929-Gal4* circuit.

**Figure supplement 4**. Characterization of *NPF* transcript levels.

Table 2. Gal4 and RNAi lines used to screen for CH503-related defects

| Stock | Expression pattern or gene targeted* | Source |
|---|---|---|
| c386a-Gal4 | EB | www.fly-trap.org |
| 2-72-Gal4 | EB | gift from U Heberlein (Janelia Farm, VA, USA) |
| 4-67-Gal4 | EB | Gift from U Heberlein |
| 9-161-Gal4 | PC, FSB, MB | Gift from U Heberlein |
| 2-13-Gal4 | interneurons, PC | Gift from U Heberlein |
| c819-Gal4 | (Pan et al., 2012) | (Pan et al., 2012) |
| c061-Gal4 | PC, FSB, MB | (Pan et al., 2012) |
| R71GO1-Gal4 | PI, VNC | (Pan et al., 2012) |
| OK107-Gal4 | MB | DGRC #106098 |
| TrH-Gal4 | 5HT cells | Bloomington #10531 |
| TH-Gal4 | DA cells | Bloomington #8848 |
| Tdc2-Gal4 | TA and OCT cells | Bloomington #9313 |
| MB247-Gal4 | MB, EB | Bloomington #50742 |
| JO-15-Gal4 | Johnston's organ | Bloomington #6753 |
| DDC-Gal4 | DA and 5HT cells | Bloomington #7010 |
| c309-Gal4 | MB, SEZ, CX, AL, PI, TG | Bloomington #6906 |
| c305-Gal4 | MB, EB, AL, glia | Bloomington #30829 |
| c305-Gal4 | MB | Bloomington #30829 |
| c205-Gal4 | FB | Bloomington #30827 |
| c161y-Gal4 | EB, FSB, PC, chordotonal organ | Bloomington #27893 |
| c107-Gal4 | EB, FSB, PC, chordotonal organ | Bloomington #30823 |
| 30Y-Gal4 | MB | Bloomington #30818 |
| CheB42a$^{\Delta 5-68}$ | CheB42a | (Park et al., 2006) |
| UAS-TbH-RNAi #1, 2 | Tyrosine β-hydroxylase | VDRC #107070, 51667 |
| UAS-OAMB-RNAi #1, 2, 3 | Mushroom body OA receptor | Bloomington #31233, 31711; VDRC #106511 |
| UAS-Cha-RNAi | Choline acetyltransferase | VDRC #20183 |
| UAS-VAChT-RNAi | Vesicular acetylcholine transporter | VDRC #40918 |
| UAS-DAT-RNAi | DA transporter | VDRC #12082 |
| UAS-TH-RNAi | Tyrosine hydroxylase | VDRC #3308 |
| UAS-Dop1R1-RNAi | DA 1-like receptor 1 | VDRC #107058 |
| UAS-Dop1R2-RNAi | DA 1-like receptor 2 | VDRC #105324 |
| UAS-VGAT-RNAi #1, 2 | Vesicular GABA transporter | VDRC #103586, 45916 |
| UAS-Tdc2-RNAi | Tyrosine decarboxylase | Bloomington #25871 |
| UAS-SERT-RNAi | 5HT transporter | VDRC #11346 |
| UAS-GABA B R1-RNAi | Metabotropic GABA-B receptor subtype1 | VDRC #101440 |
| UAS-GABA B R2-RNAi | Metabotropic GABA-B receptor subtype2 | VDRC #1785 |
| UAS-GABA B R3-RNAi #1, 2 | Metabotropic GABA-B receptor subtype3 | VDRC #108036, 50176 |
| UAS-GAD1-RNAi | Glutamic acid decarboxylase | VDRC #32344 |

*AL: antennal lobe; CX: central complex; EB: ellipsoid body; FSB: fan-shaped body; MB: mushroom body; PC: protocerebrum; PI: pars intercerebralis; SEZ: subesophageal zone; TG: thoracic ganglion; VNC: ventral nerve cord; 5HT: serotonin; DA: dopamine; OCT: octopamine; TA: tyramine.

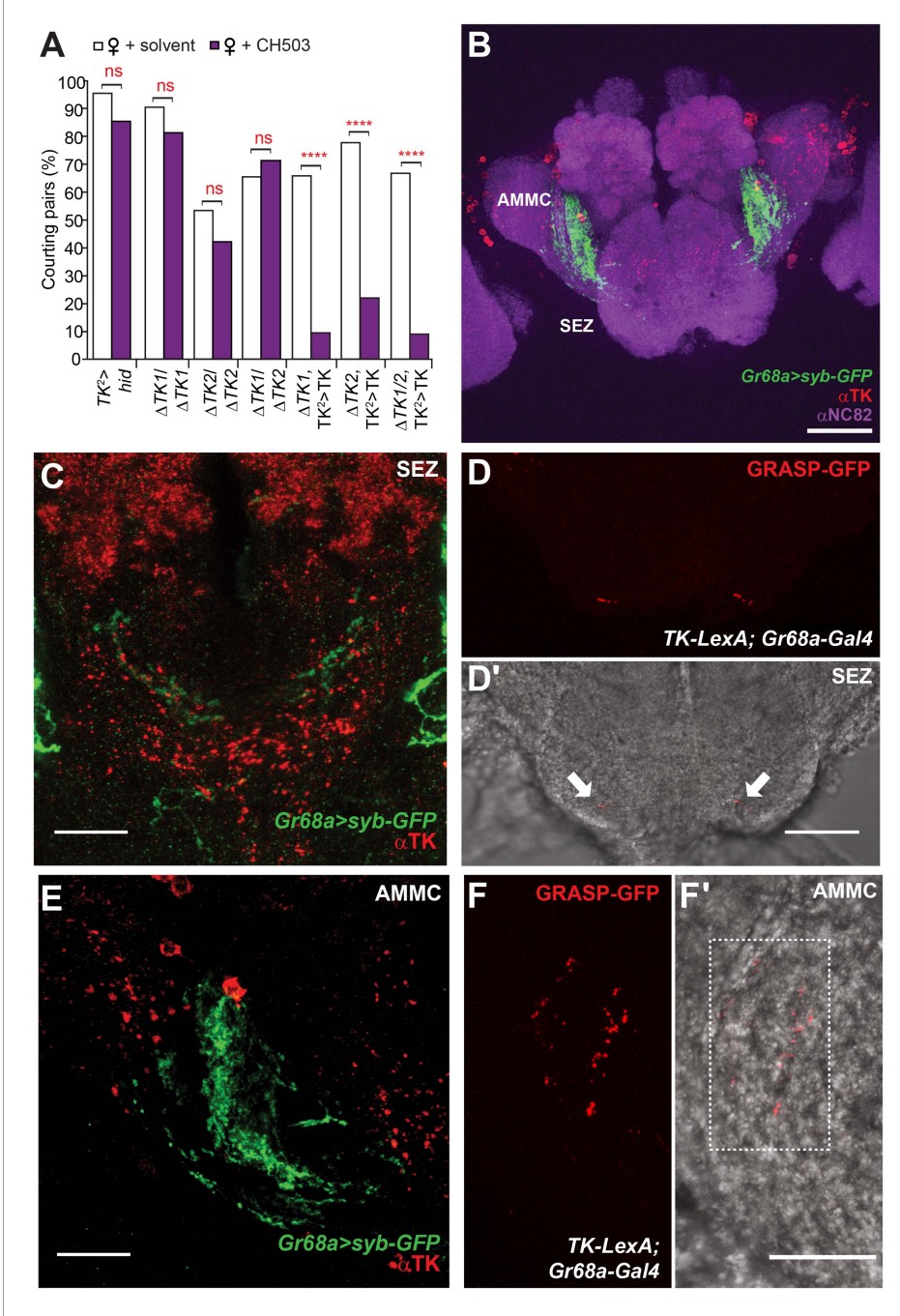

**Figure 5**. Tachykinin-expressing cells in the SEZ are a second order circuit for Gr68a neurons. (**A**) Ablation of TK-expressing circuits using two independent *Gal4* drivers (*TK²* and *TK³*) removed sensitivity to CH503. Homozygous or trans-heterozygous Δ*TK* deletion mutants also exhibit a loss of sensitivity to CH503. Rescuing *TK* expression in two different mutant backgrounds restored the behavioral response to CH503. See *Figure 5—figure supplement 2* for parental controls. N = 15–31, Fisher's exact probability test, ns: not significant, ****p < 0.0001. (**B**, **C**, **E**) TK-expressing cells are closely apposed to *Gr68a-Gal4* synaptic terminals labeled with synaptobrevin-GFP (syb-GFP) in the SEZ (**C**) and AMMC (**E**). No co-localization is observed (Pearson's coefficient: 0). Image represents a maximum intensity Z-series projection. (**D**, **D'**) Positive GRASP-GFP signal in the SEZ indicates synaptic connectivity between Gr68a neurons and TK processes. The GFP signal is overlaid on a phase-contrast image of the tissue (arrows). (**F**, **F'**) Positive GRASP-GFP signal in the AMMC indicates synaptic connectivity between Gr68a neurons and TK processes. The GFP signal is overlaid on a phase-contrast image of the tissue (rectangle). Scale bar **B**: 50 μm; all other scale bars: 20 μm.

*Figure 5. continued on next page*

*Figure 5. Continued*

The following figure supplements are available for figure 5:

**Figure supplement 1**. Tachykinin is essential for CH503 detection.

**Figure supplement 2**. Parental control lines for tachykinin mutant rescue experiments.

**Figure supplement 3**. Non-specific diffuse staining is observed in tissue from GRASP negative controls lacking the *Gr68a-Gal4* driver.

---

expressed using *Gr68a-Gal4* and *TK-LexA* drivers. By themselves, the fragments are non-fluorescent. However, close apposition of the fragments reconstitutes a functional GFP molecule and in this way, labels sites of synaptic contacts. Positive GRASP-GFP staining was apparent in the SEZ (*Figure 5D*; 6 out of 10 brains) and the AMMC (*Figure 5F*; 9 out of 10 brains), indicating connectivity. In tissue from negative controls lacking the *Gr68a-Gal4* driver, only diffuse non-specific staining is seen (4 out of 4 brains; *Figure 5—figure supplement 3*).

## TK-expressing cells are essential for gustatory pheromone detection

We next asked whether the TK-positive cells involved in the processing of CH503 overlap with the *NPF-Gal4* or *c929-Gal4* circuit. Indeed, silencing *TK* expression only within either of these *Gal4*-defined populations resulted in a reduced response to CH503, with male courtship levels indistinguishable from those of the solvent-perfumed fly (*Figure 6A,C*). Conditional suppression of *TK* only from late pupal stage onwards (using temperature-sensitive *Gal80*) also produced the same phenotype, indicating that loss of CH503 sensitivity from *TK* knockdown is not due to non-specific developmental effects (*Figure 6A*).

To examine the intersection of TK- and NPF-defined circuits, *TK²-Gal4*-labeled brains were stained with an antibody to NPF. Co-expression in 7 cells (designated NPF-TK cells) was evident in the anterior ventrolateral protocerebrum (*Figure 6B*). However, two lines of evidence indicate that these 4 cells are not sufficient for mediating courtship suppression. First, on a TK mutant background, rescuing TK expression only within the NPF circuit did not restore the courtship suppression response (*Figure 6A*). Second, the four NPF-TK cells are not labeled by *TK³-Gal4*, a second population of cells that was also found to contribute to CH503 detection (*Figure 6—figure supplement 1*).

We next attempted to rescue pheromone sensitivity by replacing TK expression only within the *c929-Gal4* circuit. In wild-type flies, co-expression is evident in 10 cells in the SEZ, designated as c929-TK cells (*Figure 6D,D'*). Rescue of TK expression only in these 10 cells restored the pheromone-response behavior in the *ΔTK2* and *ΔTK1/2* transheterozygote mutant background (*Figure 6C*). Failure to rescue in the *ΔTK1* background is likely due to a stronger mutant allele. Triple labeling of NPF and TK within the *c929* circuit revealed positive but variable co-expression between animals (*Figure 6E–G*). The majority of tissue samples contained between 1 and 3 triple labeled cells (*Figure 6F*), with one brain showing as many as 7 cells (*Figure 6G*). Taken together, these results indicate that TK release from a cluster of 8–10 cells within the SEZ is necessary to mediate the response to the gustatory pheromone CH503. In addition, *NPF-Gal4* and TK expression overlap in a small population of cells in the SEZ and protocerebrum.

## Discussion

Our work describes a neural circuit that is essential for the detection of a gustatory sex pheromone. Sensory neurons in the male foreleg that house the gustatory receptor Gr68a detect the sex pheromone CH503 and relay information to higher order neural centers via peptidergic neurons in the SEZ. The release of TK from a cluster of 8–10 cells found within the *c929*-labeled SEZ region is essential for mediating the courtship suppression response.

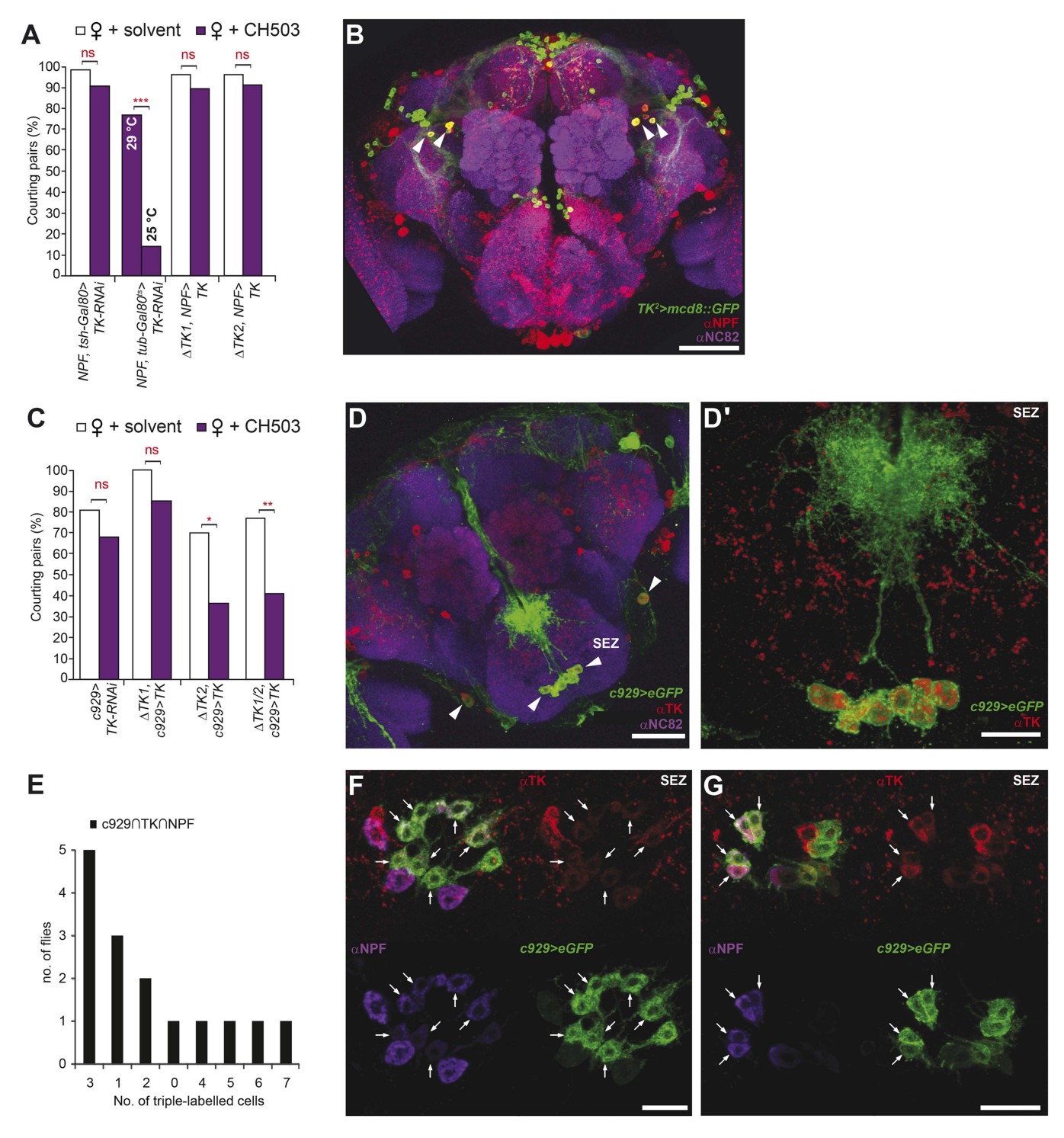

**Figure 6**. Tachykinin release within the *NPF-* and *c929*-defined circuits is required for the processing of CH503. (**A**) RNAi-mediated knockdown of *TK* only in central *NPF-Gal4* circuits abrogates the CH503-induced courtship suppression response. Conditional knockdown only from late pupal stage onwards (29°C permissive temperature, *TK-RNAi* expressed) elicits the same phenotype. At the 25°C restrictive temperature (*TK-RNAi* not expressed), flies continue to respond to the pheromone. Restoring *TK* expression only in the *NPF-Gal4* circuit is not sufficient to restore sensitivity to CH503. See ***Figure 6—figure supplement 2*** for parental controls. N = 14–23, Fisher's exact probability test, ns: not significant, ***p < 0.001. (**B**) Co-expression and co-localization of anti-NPF immunostaining with *TK²-Gal4* processes is observed only in two pairs of bilateral cells in the ventrolateral protocerebrum (indicated by arrowheads; Pearson's coefficient: 0.7). No co-expression is observed in midline cells—the apparent co-localization observed in some cells (yellow signal)
*Figure 6. continued on next page*

*Figure 6. Continued*

is due to overlapping signals from stacking different optical layers. Image represents a maximum intensity Z-series projection. Scale bar: 50 μm. (**C**) Silencing *TK* expression only in the *c929-Gal4* circuit removes CH503 sensitivity. Rescuing *TK* expression only in the *c929-Gal4* circuit restores sensitivity. N = 16–24, Fisher's exact probability test, ns: not significant, *p < 0.05. (**D**) Co-expression of anti-TK immunostaining with *c929-Gal4* GFP expression is observed in 10 cell bodies housed in the SEZ (arrowheads; **D′**). Images represent maximum intensity Z-series projections. Scale bar **D**: 50 μm; **D″**: 35 μm. (**E**) Histogram showing frequency of cells in the SEZ that are triple-labeled with anti-NPF antibody, anti-TK antibody, and *c929-Gal4* GFP expression. (**F, G**) Seven or three triple-labeled cells (arrows) in the SEZ. Images represent maximum intensity Z-series projections. Scale bar: 20 μm.

The following figure supplements are available for figure 6:

**Figure supplement 1**. The *TK³-Gal4* circuit does not co-localize with NPF.

**Figure supplement 2**. Parental control lines for tachykinin mutant rescue experiments.

## A dual sensory role for Gr68a in courtship initiation

Previous characterization of Gr68a indicated that female pheromones are likely to serve as ligands and that Gr68a-impaired males are defective in their courtship of females (*Bray and Amrein, 2003*). In contrast, our study shows that Gr68a-impaired males courted females at similar levels to control males and that one ligand for the receptor is the male sex pheromone CH503. Several factors can account for these discrepancies. First, the courtship defect of Gr68a-impaired males is only evident in larger-sized behavioral chambers (*Ejima and Griffith, 2008*). Our study used smaller chambers (∅ = 10 mm) compared to the previous study (30 mm; *Bray and Amrein, 2003*) hence masking the courtship defect. Second, Ejima and Griffith previously showed that Gr68a is needed to relay motion-related cues that stimulate courtship initiation. The dual sensory role of Gr68a is consistent with its expression pattern in males in separate mechanosensory and gustatory neuron populations (*Ejima and Griffith, 2008*) and afferent projections to the SEZ and AMMC, a relay center for auditory information. Thus, males' inability to detect motion rather than female pheromones may underlie the previously reported Gr68a-related courtship initiation defects. Taken together with the results of our current study, we propose that excitatory mechanosensory information is integrated with inhibitory CH503 detection via Gr68a neurons and both types of sensory cues shape the decision to initiate and sustain courtship. This model is consistent with our findings that activation of Gr68a-neurons using TrpA1 did not promote courtship suppression. Simultaneous activation of both courtship-promoting and suppressing channels likely resulted in an overall null response. Similar dual functioning neurons integrating chemical information with touch have been described in nematodes (*Kaplan and Horvitz, 1993*), mammalian olfactory receptors (*Grosmaitre et al., 2007*), and vertebrate nociceptor systems (*Besson and Chaouch, 1987*; *Woolf and Walters, 1991*).

## Interaction of CH503 with Gr68a

The nature of the ligand–receptor interaction is an open question in the absence of data from heterologous expression studies or structure–function analysis. Our observation that CH503 induces courtship inhibition only when detected on female cuticles has several implications for the underlying mechanisms. Potentially, activation of Gr68a receptors found on other parts of the male body (e.g., other regions of the legs or wings) is needed and occurs only when males assume courtship postures. Alternatively, concurrent detection of a co-factor residing on female cuticles might be required. Though our results show that hexane-soluble molecules on the cuticle are not needed, the possibility remains that cuticular peptides or proteins may help transduce pheromone detection. Previous work indicated that a pheromone binding protein, CheB42a, is found in the lumen surrounding Gr68a-neurons and is needed for detection of female pheromones (*Park et al., 2006*). While CheB42a mutants are still sensitive to CH503, other families of binding proteins may be used to detect female-specific signals that facilitate the activity of CH503. Lastly, tactile contact might be needed together with chemosensory stimulation. Previously, Kohatsu et al. showed that in *Drosophila*, contact between the male foreleg and female abdomen was necessary to initiate courtship (*Kohatsu et al., 2011*). Abdomen contact induced transient activity in the transmidline interneurons of the P1 neuron cluster, a designated 'command center' for courtship behavior that integrates input

from multiple sensory modalities (*Kimura et al., 2008*; *Kohatsu et al., 2011*; *von Philipsborn et al., 2011*). Information relayed from the upstream targets of Gr68a-expressing neurons could contribute to the silencing of P1 neurons and in this way, lower the probability of sustained courtship behavior. Whether both chemical and tactile cues are mediated by the same receptor or the same cell remains to be determined.

## Relevance of Gr68a neuron physiological responses to mating behavior

The majority of Gr68a neurons responded to CH503 at a 500 ng dose, an amount consistent with the behaviorally active dose. However, the quantity that is transferred to the female during courtship is approximately 60–100 ng based on semi-quantitative mass spectral analysis of recently mated females. Why is there a discrepancy between the amount that is needed for biological activity and the amount that is transferred? It could be that the presence of cVA, another anti-aphrodisiac, reduces the necessity for a large amount of CH503 or that each molecule synergizes the efficacy of the other. Additionally, recent findings indicate that male *D. melanogaster* have adapted to become less sensitive to CH503 to circumvent the courtship inhibitory response (*Ng et al., 2014*). While ~80 ng is sufficient to inhibit male courtship in other non-CH503 producing drosophilid species, *D. melanogaster* males (and males of other species that express CH503) have become less sensitive to the molecule in order to avoid chemical coercion from other males (*Ng et al., 2014*). Thus, it is not surprising that males are not sensitive to the amount of CH503 transferred to females.

## Gr68a and pickpocket neurons represent two different populations of gustatory pheromone-sensing neurons

The ion channels ppk23 and ppk25 are expressed in a specialized class of gustatory neurons that were recently shown to be involved in the detection of non-volatile, male- and female-specific cuticular hydrocarbons (*Lu et al., 2012*; *Thistle et al., 2012*; *Toda et al., 2012*; *Vijayan et al., 2014*). Though we were able to measure CH503-induced physiological activity from ppk23-expressing neurons on the proboscis and the male foreleg, the responses in the foreleg are not likely to be specific to the natural pheromone since behaviorally inert chemical analogs also caused a change in $Ca^{2+}$ flux. It is unclear as well whether the activity of ppk23 and ppk25-expressing neurons contributes to CH503-mediated courtship avoidance since males continue to respond to the pheromone (albeit more weakly) when expression of either channel is suppressed. Since ppk23 neurons co-localize with *fruitless* (*Thistle et al., 2012*; *Toda et al., 2012*), a master gene that regulates many aspects of courtship behavior (*Yamamoto and Koganezawa, 2013*), whereas Gr68a neurons do not (*Figure 3—figure supplement 6*), it is clear that Gr68a and ppk23 expression will also be in distinct populations of neurons. Based on previous observations that ppk23 neurons respond to both attractive and aversive sex pheromones, we speculate that they function as general detectors for a subset of chemosensory cues rather than encode information about valence.

## Comparison of the CH503 gustatory circuit to circuits for other tastants and pheromones

Afferent inputs from Gr-expressing sensory neurons extend to the neuromeres of the TAG and the SEZ in distinct projection patterns (*Kwon et al., 2014*). Are there distinct loci within the SEZ that differentially process food and pheromonal cues? In the olfactory system of the fly, information from fruit and pheromones appears to segregate into distinct regions of the lateral horn of the protocerebrum (*Jefferis et al., 2007*) (though some exceptions have been found, see *Ronderos et al., 2014* and *Grosjean et al., 2011*). In the SEZ, the distinctions are less clear. The Gr32a, Gr33a, and Gr39a receptors, which are predicted to detect courtship-inhibitory pheromones, overlap with bitter-sensing neurons in the labellum (*Wang et al., 2004*; *Weiss et al., 2011*) and have very similar SEZ projection patterns (*Kwon et al., 2014*). In addition, no obvious differences in receptor expression at the periphery have been found between males and females. In contrast, Gr68a expression is specialized for male forelegs, appearing in both gustatory and mechano-sensory neuron populations. Expression in females is largely restricted to cells located next to chordotonal organs (*Ejima and Griffith, 2008*). Moreover, the Gr68a neuron projection pattern to the TAG appears unique amongst the gustatory receptors, perhaps reflecting its dual role in mechanosensation and chemical detection (*Kwon et al., 2014*). Refined analysis of individual

afferent processes in the SEZ will allow us to better understand whether the quality and valence of a tastant is encoded at the level of the sensory receptor or transformed within the SEZ and higher order regions. Intriguingly, the AMMC was recently identified as a higher-order processing target for gustatory projection neurons from the SEZ which convey sweet information (*Kain and Dahanukar, 2015*). Perhaps some AMMC-projecting neurons may exist for bitter or pheromone-specific information.

## Central processing of pheromone detection by peptidergic circuits

The decision to mate involves considerable investment of resources and risk of predation (*Daly, 1978*). In this regard, the TK circuitry serves as a point of convergence for multiple, opposing physiological drives which underlie an animal's decision to mate (*Figure 7*). TK has been implicated in numerous systemic functions including lipid metabolism (*Song et al., 2014*), stress resistance (*Kahsai et al., 2010*), and modulation of aggression levels (*Asahina et al., 2014*). In addition, the mammalian homolog Substance P controls sexual behavior, stress responses, appetite, and aggression (*Argiolas and Melis, 2013*). The convergence of aggression and sex within a common neural pathway alludes to the possibility that the choice to fight or court is modulated by TK release which is, in part, regulated by external stimuli such as pheromones. Interestingly, the c929-TK cells identified in this study were shown previously to be involved in water conservation in response to desiccation (named ipc-1 and 2a; [*Kahsai et al., 2010*]), indicating that physiological stress could inform the decision to initiate and sustain courtship via TK signaling.

The NPF peptide itself does not appear to play a direct role. It is likely that the circuit defined by *NPF-Gal4* line effectively inhibits the response to CH503 due to expression in non-NPF expressing cells, some of which express TK. It is intriguing to consider that NPF-related circuits were shown recently to be important in transducing pheromone perception from ppk23-positive cells to the central brain (*Gendron et al., 2014*). Perhaps overlapping TK circuits may function as a second order circuit for ppk23 neurons.

In summary, the circuit described in this work provides a tractable model in which to study how gustatory pheromone information converges with physiological state to modulate complex social behavior. Identifying the neural substrates that respond to TK release and determining how third order neurons integrate with components for processing olfactory pheromone information will be essential in piecing together a connectome for mating behavior.

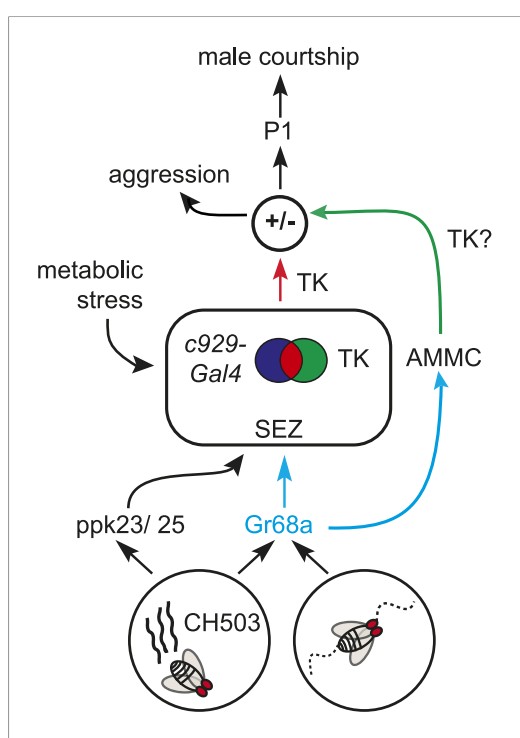

**Figure 7**. A model for gustatory pheromone perception in peripheral and central neural circuits. Gr68a neurons on the foreleg relay chemosensory and mechanosensory signals to the subesophageal zone (SEZ) and AMMC, respectively. Movement detection via Gr68a neurons contributes to the decision to court, possibly through TK signaling. The c929-TK cell cluster within the SEZ transduces information via TK release to higher order centers, potentially including the P1 courtship 'command center'. Metabolic stress responses and pheromone detection converge on the c929-TK cluster. Overall levels of TK release from this group of 8–10 cells could modulate the behavioral switch between aggression and courtship.

## Materials and methods

### Fly stocks

The following lines were used: *Or83b-Gal4* (*Larsson et al., 2004*); *Gr-Gal4* collection including *Gr68a-Gal4[2]* (*Weiss et al., 2011*); *Gr68a-Gal4[1]* (*Bray and Amrein, 2003*); *ppk23-Gal4*, *Δppk23*, and *Δppk29* (*Thistle et al., 2012*); *Voila[1]* (*Balakireva et al., 1998*); *NPF-Gal4* and *NPFR1-Gal4* (*Wu et al., 2003*); *c929-Gal4*

(*Hewes et al., 2003*); *oeno-Gal4* and *UAS-hid, stinger* (*Billeter et al., 2009*); *tsh-Gal80* (kind gift of Julie Simpson); *UAS-GCaMP5G* (*Akerboom et al., 2012*); *ΔTK1, ΔTK2*, and *UAS-TK* (*Asahina et al., 2014*); *UAS-spGFP* and *LexAop-spGFP* (*Gordon and Scott, 2009*); *TK-Gal4[1]* (#51975), *TK-Gal4[2]* (#51974), *TK-Gal4[3]* (#51973), *TK-LexA* (#54080), *UAS-mCD8:GFP*, *UAS-stinger*, *UAS-syt.eGFP*, *UAS-reaper*, *UAS-dORKΔC*, *UAS-DTI*, *UAS-Shibire[ts1]*, *UAS-dTrpA1* (Bloomington Stock Center, Indiana, USA); *UAS-Gr68a-RNAi* (13380, 13381 from VDRC, Vienna, AU) and *UAS-TK-RNAi* (103662 from VDRC). All other stocks used for screening are described in *Table 2*.

## Transgenic flies

*ΔGr68a* and *ΔGr68a*-rescue (*Gr68a[Res]*) flies were generated by ends-out homologous recombination as previously described (*Chen et al., 2011*) using the pw25-RMCE-targeting vectors and verified by PCR using primers to the vector sequence (*Weng et al., 2009*). Loss of the *Gr68a* sequence was verified by quantitative PCR. To generate *Gr68a-Gal4* and *UAS-GCaMP5*-expressing alleles in the mutant and rescue backgrounds, *Gr68a-Gal4[2]* and *UAS-GCaMP5* transgenes were re-combined onto flies with the *ΔGr68a* or *Gr68a[Res]* background and verified by labeling with *UAS-mCD8::GFP*.

## Chemical reagents

The chemical syntheses of (3$S$,11$Z$,19$Z$)-CH503, (3$R$,11$Z$,19$Z$)-CH503, ($S$)-3-Acetoxy-19-octacosen-1-ol and ($R$)-3-Acetoxy-11, 19-octacosadiyn-1-ol have previously been described (*Mori et al., 2010*; *Shikichi et al., 2013*). All other solvents and reagents were obtained from Sigma–Aldrich (St. Louis, MO, USA).

## Courtship assay

Males (5–10 days old) were isolated at the pupal stage and raised at 23°C with 60% humidity in 10 ml polypropylene vials containing 2 ml of standard cornmeal media. A decapitated virgin female target and a 5–10 days old socially naïve experimental male were placed in a courtship chamber (∅: 10 mm, height: 3 mm) and digitally recorded for 30 min. The female courtship targets were perfumed with CH503 or evaporated solvent (control) as previously described (*Yew et al., 2009*). Briefly, six female flies were placed in 1.5-ml glass vials containing 0.25, 0.5, 1, 2, or 4 µg of ($R$, $Z$, $Z$)-CH503 and vortexed three times with 20 s rest intervals. Approximately 25% of the vial contents are transferred to the flies using this method (*Billeter et al., 2009*). A single fly from each vial was tested using direct analysis in real time mass spectrometry (DART MS; [*Yew et al., 2008*, *2009*]) to check the abundance of the CH503 signal relative to other cuticular hydrocarbons.

For the tests of CH503 volatility, a 2-layer courtship chamber was constructed with the top layer containing a male and an unperfumed decapitated virgin female and the bottom layer containing filter paper overlaid with 64 µg of ($R$, $Z$, $Z$)-CH503. To prepare female fly extract, 1 or 2 flies were submerged in hexane for 10 min at room temperature, after which the solvent was removed and added to the filter paper.

The courting pairs % refers to the number of trials in which courtship was observed for longer than one minute divided by the total number of trials. Behavioral assays for perfumed and solvent-perfumed animals were performed in parallel. The courting pairs % was compared between pheromone and solvent-perfumed flies bearing the identical genetic background. Statistical analysis was performed using a Fisher's exact probability test with Yates correction (VassarStats, www.vassarstats.net). Courtship vigor and latency were calculated for the 30 min observation period and compared using a one-way ANOVA with a Tukey's post-hoc test (SPSS Statistics, IBM, USA).

## Screen for receptor and neural circuits associated with CH503 detection

Behavioral screens were performed using female targets perfumed with 83 ng/fly of ($S$, $Z$, $Z$)-CH503 or 333 ng/fly dose of ($R$, $Z$, $Z$)-CH503. Both doses were previously established as the minimum necessary for each stereoisomer to elicit significant courtship suppression (*Mori et al., 2010*). Transgenic flies were generated using the *Gal4–UAS* system to drive expression of toxin or pro-apoptotic transgenes (*UAS-Reaper*, *UAS-DTI*, and *UAS-hid, stinger*) or transgenes that interfere with synaptic transmission (*UAS-Shibire[ts1]* and *UAS–dORKΔC*). For *Gr-Gal4* lines, *UAS-Reaper* was used since *UAS-DTI* expression led to low basal courtship activity. For central circuit screening, *UAS-Shibire[ts1]* was used to avoid developmental lethality. For *Gal4* lines in which *UAS-Shibire[ts1]* expression led to paralysis,

seizures, or low courtship activity, *UAS–dORKΔC* was used as a milder form of neural inhibition. Knockdown of *Gr68a* expression was performed using an RNAi line since the use of other transgenes resulted in larval lethality. Manipulation of neurotransmitter levels was performed using RNAi to target the respective transporters, receptors, or biosynthetic enzymes. See *Table 2* for a complete list of *Gal4* and *UAS-RNAi* lines used for screening.

For experiments involving temperature-sensitive transgenes (*UAS-Shibire^{ts1}* and *UAS-dTrpA1*), flies were placed in a humidity-controlled incubator at 29°C for 2 hr prior to the assay to activate the transgene. Courtship chambers were placed on a hotplate pre-warmed to 29°C, during which time the flies were introduced into each chamber. The chamber was placed in an incubator at 29°C and the temperature was monitored throughout the assay. Control experiments were carried out in parallel at 23°C (for assays using *UAS-Shibire^{ts1}*) or 19°C (for assays using *UAS-dTrpA1*).

## Proboscis extension reflex assay

The proboscis extension reflex (PER) assay was performed as previously described (*Lacaille et al., 2007*; *Shiraiwa and Carlson, 2007*). Virgin 1-day-old male and female flies were starved for 36 hr in a vial containing tissue soaked with water. Flies were mounted with nail polish on the dorsal side onto glass slides and placed in a humidified Petri dish for at least 2 hr prior to the assay. Paper wicks coated in 20 µl of a test solution were used for bilateral stimulation of the tarsi of mounted flies. One leg was touched with a paper wick soaked in 4% sucrose (in distilled $H_2O$, wt/vol) while the second leg was simultaneously stimulated with one of the following solutions: (i) 4% sucrose, (ii) 25–100 µg/ml ($R$, $Z$, $Z$)-CH503 in hexane, (iii) 0.25–25 µg/ml ($S$, $Z$, $Z$)-CH503 in hexane, and (iv) 100 mM caffeine in $dH_2O$. Each substance was tested three times with a 2 min rest between stimulations. A response was counted as positive when the fly extended its proboscis for at least 2 of 3 stimulations. The response to each substance was compared to the response induced by 4% sucrose alone using a Fisher's exact probability test. The assays were carried out at the same time each day, and the experimenter was blind to the identity and concentration of stimulants tested in the assay.

## Immunohistochemistry

Adult *Drosophila* brains and thoracic ganglia from 6- to 10-day-old virgin flies were dissected in phosphate buffered-saline with 0.3% Triton X-100, pH 7.2 (PBST) and fixed in ice-cold 4% paraformaldehyde for 25 min. Samples were washed three times, for 15 min each in PBST, treated in a blocking solution containing PBST and 10% normal goal serum for 30 min at room temperature, and incubated in primary antibody solution. After three washes in PBST, the tissues were incubated overnight at 4°C in secondary antibody solution. Following three washes in PBST, the brains were mounted on glass slides with Vectashield mounting medium (Vector Laboratories, Burlingame, CA). Images were acquired on a Zeiss LSM 510 Meta inverted microscope equipped with 488, 543, and 633 nm lasers. For all tissues, 132 frames with a z step size of 0.46 µm were acquired. The following secondary antibodies used were anti-chicken 488 (1:500; Jackson ImmunoResearch Laboratories, West Grove, USA), anti-rabbit Cy3 (1:500; Jackson ImmunoResearch Laboratories), anti-guinea pig Cy3 (1:500; Jackson ImmunoResearch Laboratories), anti-mouse 633 (1:500; Jackson ImmunoResearch Laboratories). Image analysis was done using ImageJ software (NIH). The following primary antibodies and dilutions were used: chicken anti-GFP (1:1000; Abcam, Cambridge, UK), mouse anti-nc82a (1:50; Developmental Studies Hybridoma Bank, Iowa City, USA), mouse anti-GFP (1:1000, for GRASP; A11120, Life Technologies, NY, USA), mouse anti-GFP (1:100, for GRASP; G6539, Sigma–Aldrich), rabbit anti-NPF (1:2000; kind gift from P Shen; [*Wu et al., 2003*]), and guinea pig anti-TK (1:2000; kind gift from D Anderson; [*Asahina et al., 2014*]). The Pearson's coefficient, a measure of co-localization, was calculated using Imaris software (Bitplane AG, Zurich, Switzerland).

## Quantitative PCR analysis

For each biological replicate, RNA from 200 flash-frozen fly forelegs (for *Gr68a* experiments) or 20 flash-frozen heads (for *NPF* experiments) was extracted using TRIzol Reagent (Ambion, Austin, TX, USA) according to manufacturer's instructions. DNAse treatment was performed using TURBO DNA-free kit (Ambion) and cDNA was synthesized with SuperScript III (Invitrogen, Waltham, MA, USA) with an oligo-dT primer. Quantitative PCR was performed using CFX Connect Real-Time PCR System (Bio-Rad, Hercules, CA, USA) and SYBR Fast ABI Prism qPCR kit (Kapa Biosystems, Wilmington, MA, USA). See *Table 3*, for primer sequences and annealing temperatures.

**Table 3.** Primers used for quantitative and semi-quantitative (semi-Q) PCR experiments

|  | Forward primer (5′–3′) | Reverse primer (5′–3′) | Annealing temperature (°C) |
|---|---|---|---|
| Gr68a (qPCR) | CCAAGGTGATACCGAGGAGGAGA | TCGTGAAGAGTGCGAAAGTG | 60 |
| Gr68a (semi-Q PCR) | CCAAGGTGATACCGAGGAGA | CATTGGCCAGCAGATACTCA | 55 |
| CG6024 | CCAAGGTGATACCGAGGAGA | TCATGAAGAGTGCGAAAGTG | 60 |
| NPF | GCGAAAGAACGATGTCAACAC | TGTTGTCCATCTCGTGATTCC | 60 |
| rp49 | CCAAGGACTTCATCCGCCACC | GCGGGTGCGCTTGTTCGATCC | 55 |
| RMCE vector | GTACTGACGGACACACCGAAG | GGATCAACTACCGCCACCT | 52 |

## In vivo Ca$^{2+}$ imaging

In vivo GCaMP imaging experiments were performed on 14- to 28-day-old adults. A live fly was immobilized on a 0.17-mm coverslip with nail polish (Sally Hansen, USA). 10 µl of PBST were placed onto the tarsal segments using a pipette, after which three pre-stimulation images were acquired for a total duration of 2.4 s. For each measurement, 117 post stimulation images were acquired for a total duration of 93.6 s immediately after the addition of 10 µl of the natural stereoisomer of CH503, ($R$, $Z$, $Z$)-CH503 (final concentrations of 50 or 500 ng in PBST). Identical conditions were used for measurements using ($S$, $Z$, $Z$)-CH503. No reflux flow is used in the sample preparation. Control stimulants consisted of 10 µl of solvent or an analog of CH503: ($S$)-3-Acetoxy-19-octacosen-1-ol (50 ng, final concentration) or ($R$)-3-Acetoxy-11, 19-octacosadiyn-1-ol (500 ng, final concentration). The analogs were previously established to be behaviorally inert (*Shikichi et al., 2013*). Images were acquired on a spinning disk confocal microscope (Ti-E; Nikon Instruments, Melville, USA) equipped with a CSU-X1 scan head (Yokogawa Electric, Tokyo, JA) and either a digital sCMOS camera (ORCA-Flash4.0; Hamamatsu Photonics, Shizuoka, JA) or a cooled CCD camera (CoolSNAP HQ2; Photometrics, Tucson, USA) using a 60×/1.4 N.A. oil objective lens. A 491 nm laser was used to excite the GCaMP5 reporter. Four Z-slices with a thickness of 0.5 µm each were acquired every 800 ms, for a total of 120 frames.

To calculate the maximum change in fluorescence signal (ΔF/F), the signal density over the whole cell body was divided by the signal from an equivalent volume of an adjacent region (background). Confocal Z-stacks were analyzed using ImageJ (*Schneider et al., 2012*). For *Gr68a-Gal4*-labeled neurons, ΔF/F was calculated from single neurons. For *ppk23-Gal4*-labeled neurons on the foreleg, ΔF/F was calculated from the total signal from either two adjacent cell bodies or the base of the axon projections. Due to their close proximity to each other, some individual cell bodies could not be differentiated. In some experiments, the maximum ΔF/F occurred in projections though it could not be discerned from which cell body the projection originated. For proboscis measurements, ΔF/F represents the averaged values from 14 cells (for CH503 stimulation) or 20 cells (for PBST stimulation), measured from 5–6 flies. For all measurements, the averaged, normalized response to the stimulant vs the averaged, normalized response to control solvent was compared using a Student's *t*-test for equal or unequal variances (Vassar Stats). Comparison of variance was determined with an F-Test (Vassar Stats). Statistical power analysis was performed using G*Power 3 (*Faul et al., 2007*).

## Acknowledgements

We thank O Alekseenko, S Certel, YB Chan, YN Chiang, J Chin, A Claridge-Chang, E Dion, TW Koh, P Rothemund, and SH Ng for helpful suggestions and critical comments on the manuscript. O Alekseenko, H Amrein, D Anderson, JC Billeter, S Cohen, B Dickson, J Levine, C Montell, K Scott, P Shen, J Simpson, and L Vosshall, generously provided flies and reagents. This work was supported by the Singapore National Research Foundation (grant NRF-RF2010-06 to JYY) and the Alexander von Humboldt Foundation (JYY).

## Additional information

### Funding

| Funder | Grant reference | Author |
| --- | --- | --- |
| National Research Foundation-Prime Minister's office, Republic of Singapore | NRF-RF2010-06 | Joanne Y Yew |
| Alexander von Humboldt-Stiftung | | Joanne Y Yew |

The funders had no role in study design, data collection and interpretation, or the decision to submit the work for publication.

### Author contributions

SS, JYC, KJT, MEKC, JYY, Conception and design, Acquisition of data, Analysis and interpretation of data, Drafting or revising the article; RW, Conception and design, Acquisition of data, Analysis and interpretation of data, Drafting or revising the article, Contributed unpublished essential data or reagents; WCN, Acquisition of data, Analysis and interpretation of data, Drafting or revising the article; KM, Conception and design, Drafting or revising the article, Contributed unpublished essential data or reagents

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
