## [Decision Letter]

Thank you for sending your work entitled “The neuropeptide tachykinin is essential for pheromone detection in a gustatory neural circuit” for consideration at *eLife*. Your article has been favorably evaluated by a Senior editor and three reviewers, one of whom is a member of our Board of Reviewing Editors.

The following individuals responsible for the peer review of your submission have agreed to reveal their identity: Leslie Griffith (Reviewing editor); Aki Ejima (peer reviewer). A further reviewer remains anonymous.

The Reviewing editor and the other two reviewers discussed their comments before we reached this decision, and the Reviewing editor has assembled the following comments to help you prepare a revised submission.

This paper describes a receptor and a neuronal circuit for the detection of the male contact pheromone CH503 and provides an important advance for our understanding of the influence of gustatory pheromones on courtship behavior. For the most part the experiments are clear and the data are convincing. Several issues need to be addressed before this paper is suitable for publication.

1) Calcium imaging data. N is too low to give the reader any confidence that the conclusions drawn are robust. Simply looking at the high variability in these measurements makes it clear that N must be quite a bit higher to have the statistical power to make a conclusion. For instance, the data on the female foreleg in 3D look qualitatively similar to the male foreleg. The lack of statistical significance is likely due to low and N and it is disingenuous to suggest that this means females “do not detect” CH503. They clearly have responses that are equal in magnitude to males in T2. N has to be substantially increased and an a priori power test should be done to determine what N needs to be. Additionally, the doses/sources of pheromone used on males and females absolutely must be the same. It is completely inappropriate to conclude females do not respond if you do not use an effective dose of CH503.

2) RNAi experiments. There are a lot of RNAi lines used and their effectiveness needs to be demonstrated or references to studies that validate them provided. This is especially crucial with regard to negative results such as those shown for NPF-RNAi. Concluding that this peptide is not required based on a line that is not shown to reduce NPF immunoreactivity is not a very strong argument without the appropriate control experiment. An additional independent NPF-RNAi would also strengthen the case.

3) Circuit. The actual nature of the circuit is not clear. The authors posit a direct connection between Gr68a neurons and NFP-positive cells. Are these also TK-positive cells? Are there synaptic connections between Gr68a neurons and TK neurons? Are there functional connections? GRASP and/or functional imaging or even some better immunohistochemistry with anti-TK to show where those cells are with respect to the Gr68a terminals would be helpful along with a more precise and clear explanation of the model.

4) In vivo relevance of dosages of CH503.There needs to be at least a discussion of the levels of CH503 present on females during normal courtship and how this relates to dosages used in this study.

5) Exact nature of the CH503/Gr68a interaction. The behavioral and genetic results are consistent with a direct receptor/ligand interaction, but without heterologous expression of Gr68a and demonstration of direct effects of CH503, it is still possible that the interaction is more complex (especially in light of the requirement for it to be presented on a female to be effective). The authors need to acknowledge the potential complexity of the interaction if they cannot provide this type of evidence.

---

## [Author Response]

*1) Calcium imaging data. N is too low to give the reader any confidence that the conclusions drawn are robust*.

We thank the reviewers for raising the important issue of statistical power and sample size. For male imaging experiments with CH503, the sample size has been increased, and now ranges from 4-17 cells for each cell and each dose (see Figure 3). Sample size for each cell is indicated in the figure. In addition, we have performed statistical power tests for the experiments using 50 and 500 ng CH503. For the male trials, the sample size was sufficient to reach the commonly accepted power level of 0.8 (Cohen, 1988) for a significance level of 0.05 for almost all of the cells. There were 2 exceptions: For Cell h, 500 ng dose and Cell c, 50 ng dose. For the former, a sample size of 201 is needed. For the latter, a sample size of 47 is needed. We have added statistical power to the figure legend and note the two instances which are underpowered.

*Simply looking at the high variability in these measurements makes it clear that N must be quite a bit higher to have the statistical power to make a conclusion. For instance, the data on the female foreleg in 3D look qualitatively similar to the male foreleg. The lack of statistical significance is likely due to low and N and it is disingenuous to suggest that this means females “do not detect” CH503. They clearly have responses that are equal in magnitude to males in T2. N has to be substantially increased and an a priori power test should be done to determine what N needs to be. Additionally, the doses/sources of pheromone used on males and females absolutely must be the same. It is completely inappropriate to conclude females do not respond if you do not use an effective dose of CH503*.

In the original figure, the more potent (*S,Z,Z*)-CH503 stereoisomer was used to stimulate the female leg (Figure 3) whereas the figure for the male leg (Figure 3) showed response to the (*R,Z,Z*)-CH503 stereoisomer. The proper comparison should have been between Figure 3 and Figure 3—figure supplement 2 in the original manuscript. When the responses of males and females to (*S,Z,Z*)-CH503 are compared, it is evident that response from males is 3-5X higher than that of females.

Nevertheless, we acknowledge that the data are presented in a confusing way and that it is more appropriate to compare male to female response with the natural (*R,Z,Z*)-CH503 isomer. We have now replaced the previous Figure 3 with data from female imaging experiments using 2 doses of (*R,Z,Z*)-CH503 at 50 and 500 ng. The sample size for each cell from both male and female experiments has been increased. Statistical power of at least 0.8 is achieved for most of the cells. For Cells j and s, we would have needed at least n=100. We acknowledge that the findings for cells j and s are underpowered in the figure legend.

With respect to the female response to the pheromone, nowhere in the manuscript do we state that females “do not detect CH503”. Indeed, we fully concur that it would be disingenuous to make such a strong statement given the limited sensitivity of the PER and physiological assays. Rather, we are careful to state that females “do not exhibit a change in PER” and that it is possible females “do not perceive CH503 as an aversive tastant”. With respect to calcium imaging, we state that “In females, the responses to the pheromone from the forelegs were indistinguishable from that of the solvent control”. We note that in the text and figure legend that in the case of Cells j and s, this might be due to lack of adequate sample size. Given that females do not exhibit significant responses using both a behavioral and physiological assay, we can conclude that females do not respond robustly to CH503 in the context of these experiments and within the limits of these assays.

*2) RNAi experiments. There are a lot of RNAi lines used and their effectiveness needs to be demonstrated or references to studies that validate them provided. This is especially crucial with regard to negative results such as those shown for NPF-RNAi. Concluding that this peptide is not required based on a line that is not shown to reduce NPF immunoreactivity is not a very strong argument without the appropriate control experiment. An additional independent NPF-RNAi would also strengthen the case*.

We have measured *NPF* transcript levels in controls and RNAi knockdown animals (*elav>NPF*^*RNAi*^*)* using quantitative PCR and find a 5.3-fold decrease of transcript levels in the knockdown animals (Figure 4—figure supplement 4). We cannot exclude the possibility that a stronger knockdown might be needed to elicit a phenotype. However, since a 1.5 – 2 fold change of mRNA levels is considered well-repressed in the literature, it appears that the RNAi line is effective in inhibiting *NPF* transcription. Taken together with our findings that knockdown of the NPF receptor also had no effect on CH503 sensitivity, it is not likely that the NPF peptide is a major contributor to processing of CH503 information.

We thank the reviewers for indicating that a reduction in immunoreactivity might not be observable. It is interesting and helpful to learn this. We performed qPCR rather than immunostaining to characterize the RNAi effect. While mRNA levels may not necessarily correlate with peptide levels, quantification by qPCR provides greater target specificity than an antibody. In addition, by quantifying transcript from whole brain tissue, we avoid potential artifacts and inaccuracies that can result from the method of counting immunopositive cells (e.g., different numbers are obtained depending on whether whole brain or slices are used). Immunostaining may also not be sensitive enough to allow slight but significant differences in signal intensity to be detected.

With respect to the efficacy of Gr68a and TK RNAi lines, we use multiple genetic methods to confirm the phenotype initially observed with RNAi. For Gr68a, we regenerated a mutant by homologous recombination and show that the mutant phenocopies the Gr68a>RNAi flies in courtship behavior (Figure 2), PER assay (Figure 2), and physiological measures (Figure 2). Furthermore, the behavioral phenotype is rescued upon replacement of the *Gr68a* gene in the mutant (Figure 2). Regarding tachykinin, the TK>RNAi phenotype is phenocopied with TK>hid flies (in which TK-expressing cells are ablated), and 2 TK mutant alleles and the transheterozygote (Figure 5). The phenotype is rescued in the TK mutant alleles upon ectopic expression of the *TK* gene.

With respect to small neurotransmitter RNAi lines used in the *c929* screen (Figure 4—figure supplement 3), we now include data showing results from behavioral assays using a second RNAi line or other RNAi lines that target the recycling, transport, or receptor of each small neurotransmitter system:

A)
*Octopamine*: 2 TbH-RNAi lines and 3 octopamine mushroom body receptor (OAMB-RNAi lines).

B)*Acetylcholine*: vesicular acetylcholine transporter (VAChT-RNAi) and choline acetyltransferase (Cha-RNAi).

C) *Serotonin*: neural-specific tyrosine decarboxylase (Tdc2-RNAi) and serotonin transporter (SERT-RNAi).

D) *GABA*: metabotropic GABA receptor subtype 1, 2, and 3-RNAi (2 lines used for subtype 3), vesicular GABA transporter (VGAT-RNAi, 2 different lines) and glutamate decarboxylase (Gad1-RNAi).

E) *Dopamine*: 2 Dopamine 1-like receptor-RNAi (Dop1R1- and Dop1R2-RNAi), dopamine transporter (DAT-RNAi) and tyrosine hydroxylase (TH-RNAi).

By using multiple RNAi lines, overlapping cell populations relevant to each neurotransmitter system are targeted. All results were consistently negative. We cannot eliminate the possibility that some of these neurotransmitters may play a role in CH503 detection. It is well known that co-expression of peptides and aminergic transmitters within the same cells play an important role in many circuits. However, within the scope of the paper, we focused on two peptidergic systems whose manipulation yielded the most robust phenotype. Synergistic actions between small molecules and peptides within behavioral circuits is an important and fascinating topic for future studies.

*3) Circuit. The actual nature of the circuit is not clear. The authors posit a direct connection between Gr68a neurons and NFP-positive cells. Are these also TK-positive cells? Are there synaptic connections between Gr68a neurons and TK neurons? Are there functional connections? GRASP and/or functional imaging or even some better immunohistochemistry with anti-TK to show where those cells are with respect to the Gr68a terminals would be helpful along with a more precise and clear explanation of the model*.

Thank you for these suggestions. We have attempted to clarify the nature of the circuit by adding the following data:

A) Immunocytochemistry showing TK expression in close proximity to Gr68a synaptic terminals in the SEZ (Figure 5).

B) GRASP experiments using *TK-LexA* and *Gr68a-Gal4* show reconstituted GFP (and hence, synaptic connectivity) in the SEZ (Figure 5).

C) Triple-labeling experiments showing the overlap of NPF, TK, and the *c929-Gal4* circuit in the SEZ (Figure 6). We examined 15 brains and noted fly-to-fly variation. 3-5 triple labeled cells were most commonly found. We provide images and histograms depicting the variation in labeling.

D) Behavioral experiments showing that rescue of TK using the *c929* driver in a TK mutant background is sufficient to rescue the pheromone response. TK and *c929* overlap in 8-10 cells in the SEZ, several of which are also NPF-positive. (Figure 6).

Taken together, TK appears to be the primary circuit through which information from Gr68a is relayed to the central brain. Interestingly, the TK circuit overlaps with *NPF-Gal4* in the SEZ and protocerebrum. This overlap would explain why inhibition of the NPF circuit (but not NPF peptide) also interferes with CH503 pheromone perception.

*4) In vivo relevance of dosages of CH503.There needs to be at least a discussion of the levels of CH503 present on females during normal courtship and how this relates to dosages used in this study*.

The CH503 dose that is behaviorally active matches the dose that activates 6/9 of the Gr68a neurons. However, the amount that is transferred to the female during courtship is approx. 60-100 ng based on semi-quantitative mass spectral analysis of recently mated females. Why is there a discrepancy between the amount that is needed for biological activity and the amount that is transferred? It could be that the presence of cVA, another anti-aphrodisiac, reduces the necessity for a large amount of 503 or that each molecule synergizes the efficacy of the other. Additionally, recent work has shown that male *D. melanogaster* have adapted to become less sensitive to CH503 to circumvent the courtship inhibitory response (Ng et al., 2014a). In this paper, it was shown that ∼80 ng of CH503 was sufficient for inhibition in other non-503 producing drosophilid species. However, *D. melanogaster* males (and males of other species that express CH503) have become less sensitive to the molecule, probably as a strategy to avoid chemical coercion by rival males. Thus, it is not surprising that males are not sensitive to the amount of CH503 found on females. We have added a discussion on the relevance of the dosages (in the subsection headed “Interaction of CH503 with Gr68a”).

*5) Exact nature of the CH503/Gr68a interaction. The behavioral and genetic results are consistent with a direct receptor/ligand interaction, but without heterologous expression of Gr68a and demonstration of direct effects of CH503, it is still possible that the interaction is more complex (especially in light of the requirement for it to be presented on a female to be effective). The authors need to acknowledge the potential complexity of the interaction if they cannot provide this type of evidence*.

We add a brief discussion that addresses the necessity of a secondary cue derived from female cuticles or a lipid binding protein, which has been shown for other pheromone receptors (in the subsection headed “A dual sensory role for Gr68a in courtship initiation”). We speculate also that mechano- and chemosensory cues are detected by the same Gr68a cells. Exploring the nature of the ligand-receptor interaction and assessing the response of Gr68 neurons to mechanosensory stimulation will be an important future direction. However, given that no crystal structure is available for gustatory receptors nor successful heterologous expression (to our knowledge), our speculation on the nature of the ligand-receptor interaction is limited.